# ON THE EMERGENCE OF INDUCTION HEADS FOR IN-CONTEXT LEARNING

## ABSTRACT

Transformers have become the dominant architecture for natural language processing. Part of their success is owed to a remarkable capability known as *in-context learning* (ICL): they can acquire and apply novel associations solely from their input context, without any updates to their weights. In this work, we study the emergence of *induction heads*, a previously identified mechanism in two-layer transformers that is particularly important for in-context learning. We uncover a relatively simple and interpretable structure of the weight matrices implementing the induction head. We theoretically explain the origin of this structure using a minimal ICL task formulation and a modified transformer architecture. We give a formal proof that the training dynamics remain constrained to a 19-dimensional subspace of the parameter space. Empirically, we validate this constraint while observing that only 3 dimensions account for the emergence of an induction head. By further studying the training dynamics inside this 3-dimensional subspace, we find that the time until the emergence of an induction head follows a tight asymptotic bound that is quadratic in the input context length.

## 1 INTRODUCTION

How does intelligence emerge from *gradient descent*? Large language models (LLMs) have achieved highly advanced reasoning abilities, yet we still lack a principled account of how complex reasoning behaviors emerge from this simple learning rule. Understanding the inner workings of LLMs is an important avenue towards developing novel AI systems with increased reliability and efficiency.

LLMs possess a remarkable ability known as *in-context learning* (ICL). A well-trained language model can learn and apply novel associations from their input context, without additional parameter updates (Brown et al., 2020). This is in stark contrast to traditional *in-weights learning*, where novel associations are directly encoded into the model weights.

Previous work by Olsson et al. (2022) traces back the majority of transformers' ICL capabilities to a learned mechanism termed *induction head*: a pair of two consecutive attention heads that implement a simple but powerful copying rule $[\,\ldots, A, B, \ldots, A\,] \to B$. Empirical work has shown that the formation of induction heads co-occurs with a sharp decrease in the training loss and an increase in ICL accuracy (Olsson et al., 2022; Reddy, 2023). This motivates the question of the current study:

*How do induction heads emerge during training?*

While a number of theoretical studies have established the emergence of induction heads using specific staged learning algorithms Nichani et al. (2024a); Bietti et al. (2024), the precise learning dynamics during standard training remain elusive. To answer this question, we study the training dynamics of an autoregressive two-layer transformer using a minimal ICL task formulation (defined in §3) and simplified architecture. We show that in the proposed setup, only 19 dimensions of parameter space have non-zero gradients and therefore govern the entire learning trajectory. Then, we empirically show how only 3 dimensions of the parameter space are needed to form an induction head. In this reduced and interpretable parameter space, we explicitly study the dynamics of the three pseudo-parameters and analyze the formation of induction heads.

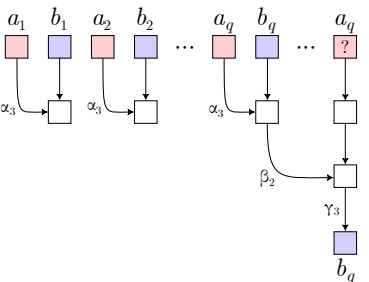 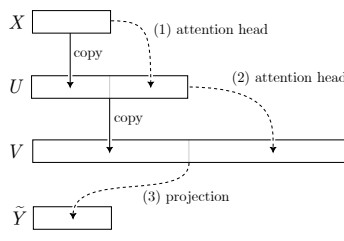

Figure 1: *Left:* an induction head solving the *in-context learning* (ICL) task. Given a series of item-label pairs, the model predicts the correct label for a query item. The first attention head retrieves the corresponding item for each label, enabling the second attention head to retrieve the correct label. Each path is modulated by one pseudo-parameter ($\alpha_3$, $\beta_2$, or $\gamma_3$). *Right:* our minimal transformer architecture. We use two attention-only layers and a linear layer. We disentangle the attention layers by concatenating the inputs and outputs, rather than adding them together.

Concretely, our contributions are as follows:

1. We train and interpret a standard attention-only transformer on an ICL task (§2). We find a relatively simple and highly interpretable **description of the weight matrices** that implement and induction head.

2. Using a minimal ICL formulation, we give a formal proof that training dynamics induce a simplified structure of the weights (§4). The evolution of model weights stays within a 19-dimensional subspace of the entire parameter space, regardless of model or task size. We index this subspace by introducing **19 pseudo-parameters**.

3. We empirically find that only **3 pseudo-parameters** are learned at the end of training, corresponding exactly to an induction head (§5). We also find that the emergence of the 3 parameters is *self-contained*, unaided by the presence of the other 16 parameters.

4. We theoretically study the training dynamics of the induction head, assuming that only the 3 parameters are learnable (§6). We prove that the 3 parameters always emerge in a specific sequence. We also prove asymptotic bounds for the emergence time for each parameter in terms of the context length, as well as a **tight bound on the total emergence time**.

Finally, we also provide empirical validation for our theoretical results.

## 2 INDUCTION HEADS

*Induction heads* are attention heads that implement a simple but powerful algorithm. Given a prompt of the form $[\,\ldots, A, B, \ldots, A\,]$, an induction head predicts the token which follows the previous occurrence of $A$, in this case being $B$. Note that induction heads are not a modified type of attention head, but rather a mechanism learned by regular attention heads during standard training.

Induction heads are composed of two attention layers. The first attention layer retrieves the value of $A$ into $B$ by attending to the previous token using positional embeddings. The newly obtained value enables the second attention layer to retrieve $B$ from the second occurrence of $A$. Note that two layers are necessary to solve the task since $B$ and the second $A$ initially have no shared information.

### 2.1 SETUP

In order to understand how induction heads are implemented, we train an autoregressive transformer following the recipe of Vaswani et al. (2017). We train the model using synthetic data to predict the label of a query item based on the preceding item-label pairs, as depicted in Fig. 1 (left). We use only two attention-only layers with one attention head per layer. We remove MLPs since they are neither necessary nor useful for the task at hand. We specify the full training details in App. E.

**Notation.** Our model has token embeddings $\boldsymbol{E} \in \mathbb{R}^{D \times N_E}$ and positional embeddings $\boldsymbol{P} \in \mathbb{R}^{D \times N_P}$. The layer $l \in \{1, 2\}$ has the query, key, value, and output matrices $\boldsymbol{W}_Q^l, \boldsymbol{W}_K^l, \boldsymbol{W}_V^l \in \mathbb{R}^{D_H \times D}$, and $\boldsymbol{W}_O^l \in \mathbb{R}^{D \times D_H}$, respectively. A final linear output layer $\boldsymbol{W}_o \in \mathbb{R}^{N_E \times D}$ is applied. We denote the embedding of token $i$ as $\boldsymbol{e}_i \in \mathbb{R}^D$ and the embedding of position $i$ as $\boldsymbol{p}_i \in \mathbb{R}^D$. Our model is configured with $D = D_H = 2048$ and $N_E = N_P = 32$.

## 2.2 WEIGHT MATRIX STRUCTURE

There are only 4 sub-spaces of the residual stream that are ever activated. First, there is the space spanned by the initial token and positional embeddings, $\boldsymbol{e}_i$ and $\boldsymbol{p}_i$. Second, there is the space where the first head writes the retrieved embeddings, $\boldsymbol{W}_O^1 \boldsymbol{W}_V^1 \boldsymbol{e}_i$ and $\boldsymbol{W}_O^1 \boldsymbol{W}_V^1 \boldsymbol{p}_i$. Third, there is the space where the second head writes the retrieved embeddings, $\boldsymbol{W}_O^2 \boldsymbol{W}_V^2 \boldsymbol{e}_i$ and $\boldsymbol{W}_O^2 \boldsymbol{W}_V^2 \boldsymbol{p}_i$. Finally, the second head could retrieve the output of the first head, creating a fourth subspace spanned by $\boldsymbol{W}_O^2 \boldsymbol{W}_V^2 \boldsymbol{W}_O^1 \boldsymbol{W}_V^1 \boldsymbol{e}_i$ and $\boldsymbol{W}_O^2 \boldsymbol{W}_V^2 \boldsymbol{W}_O^1 \boldsymbol{W}_V^1 \boldsymbol{p}_i$.

Since there are $N_E$ tokens and $N_P$ positions, each of the four subspaces will have $N_E + N_P$ dimensions. Moreover, each subspace is highly interpretable, as it can be indexed directly by the corresponding token or positional embedding. Therefore, the residual stream of our attention-only model always remains constrained to a highly interpretable $4(N_E + N_P)$-dimensional subspace.

Using these intepretable directions, we can understand the mechanism performed by each layer. For example, $\boldsymbol{p}_i^\intercal \boldsymbol{W}_K^{1\intercal} \boldsymbol{W}_Q^1 \boldsymbol{p}_j$ corresponds exactly to the attention score paid by position $i$ to position $j$ during the first layer. In Fig. 2, we visualize the key-query matrix products and final output matrix, indexed by these highly interpretable dimensions. Note that this picture is a complete description of the behavior of the model.

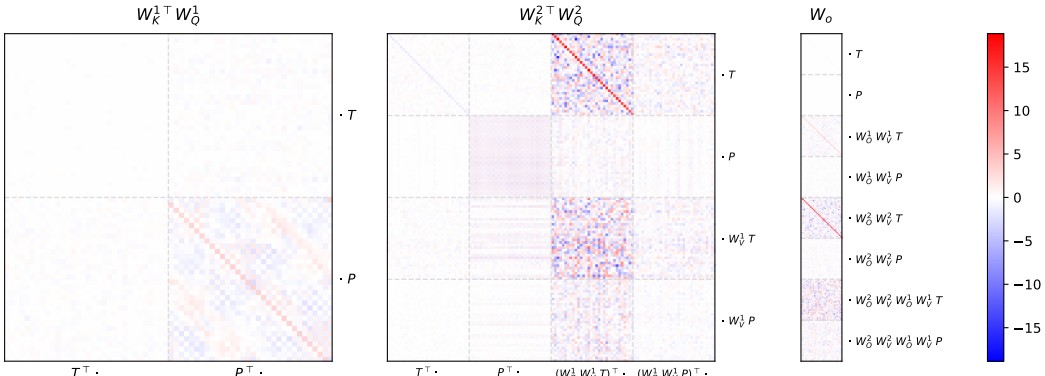

Figure 2: The complete behavior of a two-layer attention-only transformer can be understood using a highly interpretable transformation of key-query matrix products and output layer. Dots $\cdot$ denote matrix multiplication. For example, the bottom-right block of the left figure, $\boldsymbol{P}^\intercal \boldsymbol{W}_K^{1\intercal} \boldsymbol{W}_Q^1 \boldsymbol{P}$, is dominated by the subdiagonal, establishing that each position attends to the previous position during the first layer. Some noise is present due to the random initialization and stochastic gradient descent.

## 2.3 INDUCTION HEAD MECHANISM

In Fig. 2, we can see that our weights have a relatively simple and interpretable structure. Each layer is dominated by a diagonal or subdiagonal within a single block. The first layer attends to the previous position. The second layer attends to the token retrieved by the first layer. The final layer output the token retrieved by the second layer. This clarifies the structure of the weight matrices that underlie the induction head mechanism.

## 3 MINIMAL FORMULATION

In order to understand the emergence of induction heads, we study the training dynamics in a minimal formulation. Inspired by the results of the previous section, we propose a simplified, but equally powerful, transformer architecture with a *disentangled* residual stream (Friedman et al., 2023).

### 3.1 ARCHITECTURE

We use a transformer with two single-head attention-only layers followed by a linear layer. For the attention layers, we use a merged key-query matrix and no projection layer, directly concatenating the attention output to the existing residual stream:

$$\boldsymbol{H}_1 = \Big[\, \boldsymbol{X} \,\Big|\, \sigma\Big(\boldsymbol{X}\boldsymbol{W}^{(1)}\boldsymbol{X}^\intercal\Big)\boldsymbol{X} \,\Big], \quad \boldsymbol{H}_2 = \Big[\, \boldsymbol{U} \,\Big|\, \sigma\Big(\boldsymbol{H}_1\boldsymbol{W}^{(2)}\boldsymbol{H}_1^\intercal\Big)\boldsymbol{H}_1 \,\Big], \quad \widetilde{\boldsymbol{Y}} = \boldsymbol{H}_2\boldsymbol{W}^{(3)}, \tag{1}$$

where $[\,\cdot\,|\,\cdot\,]$ denotes matrix concatenation, $\sigma$ to denotes the softmax function with autoregressive masking. $\boldsymbol{W}^{(1)} \in \mathbb{R}^{2D \times 2D}$, $\boldsymbol{W}^{(2)} \in \mathbb{R}^{4D \times 4D}$, $\boldsymbol{W}^{(3)} \in \mathbb{R}^{8D \times D}$ are the learnable weights and $\boldsymbol{H}_1 \in \mathbb{R}^{(2N+1) \times 4D}$, $\boldsymbol{H}_2 \in \mathbb{R}^{(2N+1) \times 8D}$, $\widetilde{\boldsymbol{Y}} \in \mathbb{R}^{(2N+1) \times D}$ denote the activations and final output.

Although not used in practice due to computational overhead, merged key-query matrices are commonly used in theoretical works (Edelman et al., 2024; Nichani et al., 2024a). MLPs are neither necessary nor useful for the task at hand. The disentangled residual is equivalent to a very large residual dimension, where all activations become almost orthogonal.

### 3.2 DATA DISTRIBUTION

We use a common ICL task that requires labeling an item based on a list of $N$ item-label pairs (Chan et al., 2022; Reddy, 2023; Hochreiter et al., 2001). The $i^{\text{th}}$ pair consists of an item $\boldsymbol{a}_i \in \mathbb{R}^D$ and a label $\boldsymbol{b}_i \in \mathbb{R}^D$ with dimensionality $D \in \mathbb{N}$. We ask the model to predict the label for one of the items $\boldsymbol{a}_q$ where $q \in \{1, \ldots, N\}$.

We annotate each item with a positional embedding $\boldsymbol{p}_i \in \mathbb{R}^D$ and each label with the rotated positional embedding $\boldsymbol{M}\boldsymbol{p}_i$, where $\boldsymbol{M} \in \mathbb{R}^{D \times D}$. The rotation is fixed before training begins to create a learnable correlation similar to a sinusoidal embedding (Vaswani et al., 2017). This enables the attention mechanism to connect the corresponding items and labels. We do not use any positional embedding for the query item.

Assuming that $D$ is even, we use

$$\boldsymbol{M} = \left[\begin{array}{c|c} \boldsymbol{0}_{(D/2) \times (D/2)} & \boldsymbol{I}_{D/2} \\ \hline \boldsymbol{I}_{D/2} & \boldsymbol{0}_{(D/2) \times (D/2)} \end{array}\right], \tag{2}$$

where $\boldsymbol{I}_{D/2} \in \mathbb{R}^{(D/2) \times (D/2)}$ is the identity matrix.

We concatenate items and labels with their positional embeddings to obtain our data:

$$\boldsymbol{X}_{2i-1,:} = \big[\, \boldsymbol{a}_i^\intercal \,|\, \boldsymbol{p}_i^\intercal \,\big]^\intercal \qquad \boldsymbol{X}_{2i,:} = \big[\, \boldsymbol{b}_i^\intercal \,|\, \boldsymbol{p}_i^\intercal \boldsymbol{M} \,\big]^\intercal \qquad \forall i \in \{1, \ldots, N\} \tag{3}$$

$$\boldsymbol{X}_{2N+1,:} = \big[\, \boldsymbol{a}_q^\intercal \,|\, \boldsymbol{0} \,\big]^\intercal \qquad \boldsymbol{y} = \boldsymbol{b}_q \qquad q \in \{1, \ldots, N\} \tag{4}$$

where $\boldsymbol{X} \in \mathbb{R}^{(2N+1) \times 2D}$, $\boldsymbol{y} \in \mathbb{R}^D$, and $[\,\cdot\,|\,\cdot\,]$ denotes concatenation.

We assume a *lexinvariant* language model (Huang et al., 2023) where items, labels, and positional embeddings are independent and identically distributed. For our theoretical results, we introduce additional assumptions on the distribution of items, labels, and positional embeddings, as needed.

Only for our experiments, we sample $q \sim \text{unif}\{1, N\}$, and we sample items, labels, and positional embeddings from a multivariate Gaussian:

$$(\boldsymbol{a}_i)_j \sim \mathcal{N}(0, 1), \qquad (\boldsymbol{b}_i)_j \sim \mathcal{N}(0, 1), \qquad (\boldsymbol{p}_i)_j \sim \mathcal{N}(0, 1), \tag{5}$$

for all $i \in \{1, \ldots, N\}$ and $j \in \{1, \ldots, D\}$.

We train our model with mean-squared error loss $\mathcal{L} = \|\, \boldsymbol{y} - \tilde{\boldsymbol{y}} \,\|^2$ using only the output of the query item located at the last position, i.e. $\tilde{\boldsymbol{y}} = \widetilde{\boldsymbol{Y}}_{2N+1,:}$.

# 4 TRAINING DYNAMICS

Our model has a total of $28D^2$ parameters, which gives a total parameter space

$$\left[ \ \mathrm{vec}(\boldsymbol{W}^{(1)})^\intercal \ \middle| \ \mathrm{vec}(\boldsymbol{W}^{(2)})^\intercal \ \middle| \ \mathrm{vec}(\boldsymbol{W}^{(3)})^\intercal \ \right]^\intercal \ \in \ \mathbb{R}^{28D^2}$$

However, as we show below, the training dynamics on our data distribution remain constrained to a 19-dimensional subspace that we index using 19 pseudo-parameters. Our theoretical result is based on the following assumptions:

**Assumption 1.** *Zero Initialization. We assume our neural network is initialized with all weights having value zero, i.e. $\boldsymbol{W}^{(1)} = \boldsymbol{0}$, $\boldsymbol{W}^{(2)} = \boldsymbol{0}$, and $\boldsymbol{W}^{(3)} = \boldsymbol{0}$.*

The zero initialization is commonly used in theoretical works (Nichani et al., 2024a; Edelman et al., 2024), being motivated it as a reasonable approximation for small random initializations.

**Assumption 2.** *Population Loss. We assume the network is trained with gradient descent over the entire data distribution at every step:*

$$\boldsymbol{W}^{(k)} \ \leftarrow \ \boldsymbol{W}^{(k)} \ - \lambda \, \mathbb{E}\left[\frac{\partial \mathcal{L}}{\partial \boldsymbol{W}^{(k)}}\right],$$

*where $\lambda > 0$ is the learning rate.*

**Assumption 3.** *Isotropic Data. We assume that the data distribution is invariant to orthogonal transformations of items, labels, and positional embeddings:*

$$f\big(\{\boldsymbol{a}_i\}, \{\boldsymbol{b}_i\}, \{\boldsymbol{p}_i\}, q\big) = f\big(\{\boldsymbol{E}\boldsymbol{a}_i\}, \{\boldsymbol{E}\boldsymbol{b}_i\}, \{\boldsymbol{p}_i\}, q\big) = f\big(\{\boldsymbol{a}_i\}, \{\boldsymbol{b}_i\}, \{\boldsymbol{E}\boldsymbol{p}_i\}, q\big),$$

*for any orthogonal matrix $\boldsymbol{E} \in \mathbb{R}^{D \times D}$, where $f\big(\{\boldsymbol{a}_i\}, \{\boldsymbol{b}_i\}, \{\boldsymbol{p}_i\}, q\big)$ is the probability density over the items, labels, positional embeddings, and query index.*

Note that this assumption is weaker than, for example, assuming a normal distribution, since a normal distribution is isotropic.

Under these assumptions, we are able to establish that weight matrices learn the following:

**Theorem 1.** *Assume that we train a disentangled transformer from zero initialization with population loss on isotropic data on our ICL task. Then, the weight matrices will have the following structure throughout the entire training process:*

$$\boldsymbol{W}^{(1)} \ = \ \left[ \begin{array}{c|c} \boldsymbol{\alpha}_1 \boldsymbol{I} & \boldsymbol{0} \\ \hline \boldsymbol{0} & \boldsymbol{\alpha}_2 \boldsymbol{I} + \boldsymbol{\alpha}_3 \boldsymbol{M} \end{array} \right] \tag{6}$$

$$\boldsymbol{W}^{(2)} \ = \ \left[ \begin{array}{c|c|c|c} \boldsymbol{\beta}_1 \boldsymbol{I} & \boldsymbol{0} & \boldsymbol{\beta}_2 \boldsymbol{I} & \boldsymbol{0} \\ \hline \boldsymbol{0} & \boldsymbol{\beta}_3 \boldsymbol{I} + \boldsymbol{\beta}_4 \boldsymbol{M} & \boldsymbol{0} & \boldsymbol{\beta}_5 \boldsymbol{I} + \boldsymbol{\beta}_6 \boldsymbol{M} \\ \hline \boldsymbol{\beta}_7 \boldsymbol{I} & \boldsymbol{0} & \boldsymbol{\beta}_8 \boldsymbol{I} & \boldsymbol{0} \\ \hline \boldsymbol{0} & \boldsymbol{\beta}_9 \boldsymbol{I} + \boldsymbol{\beta}_{10} \boldsymbol{M} & \boldsymbol{0} & \boldsymbol{\beta}_{11} \boldsymbol{I} + \boldsymbol{\beta}_{12} \boldsymbol{M} \end{array} \right] \tag{7}$$

$$\boldsymbol{W}^{(3)} \ = \ \left[ \ \boldsymbol{\gamma}_1 \boldsymbol{I} \ \middle| \ \boldsymbol{0} \ \middle| \ \boldsymbol{\gamma}_2 \boldsymbol{I} \ \middle| \ \boldsymbol{0} \ \middle| \ \boldsymbol{\gamma}_3 \boldsymbol{I} \ \middle| \ \boldsymbol{0} \ \middle| \ \boldsymbol{\gamma}_4 \boldsymbol{I} \ \middle| \ \boldsymbol{0} \ \right]^\intercal, \tag{8}$$

*where we collect the parameters of each weight matrix in three vectors $\boldsymbol{\alpha} \in \mathbb{R}^3$, $\boldsymbol{\beta} \in \mathbb{R}^{12}$ and $\boldsymbol{\gamma} \in \mathbb{R}^4$ that vary throughout training.*

*Proof Sketch.* We give an inductive proof by showing that, if weights have the above structure, then their gradients also have the same structure. Since the zero initialization fits the structure, this ensures that the structure is preserved during training.

To prove the structure of the gradient, we apply a carefully chosen rotation to the entire data distribution. Since the data distribution is isotropic, the rotation will not change the data distribution, so the expected gradient will also remain unchanged.

However, we are also able to show that our rotation induces a specific similarity transformation of the gradient:

$$\mathbb{E}\left[\frac{\partial \mathcal{L}}{\partial \boldsymbol{W}_{ij}^{(k)}}\right] \ = \ F \, \mathbb{E}\left[\frac{\partial \mathcal{L}}{\partial \boldsymbol{W}_{ij}^{(k)}}\right] F^\intercal, \tag{9}$$

where $F$ is an orthogonal or block-orthogonal matrix and $\boldsymbol{W}_{ij}^{(k)}$ is a block of a weight matrix. From this, we are able to show that the expected gradient must have the desired structure. We give the full proof in App. A. □

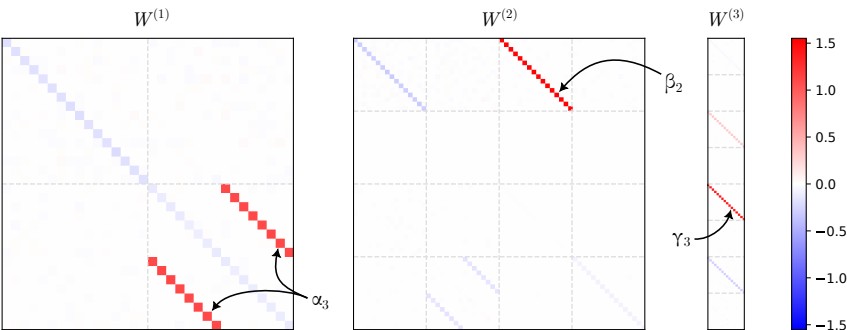

Figure 3: Weights at the end of standard training have the theoretically predicted structure.

**Empirical Validation** In Fig. 3, we confirm our theoretical result by visualizing the weights at the end of training with stochastic gradient descent. Full training details in App. C.

## 5 EMERGENCE OF INDUCTION HEADS

We now proceed to studying the evolution of these 19 pseudo-parameters during training. By observing or ablating specific parameters, we are able to test two hypotheses regarding the emergence of induction heads.

**Hypothesis 1** (due to Olsson et al. (2022)). ***Induction Head Phase Transition.*** *Reaching low training loss on our ICL task coincides with the emergence of an induction head, as defined in §2.*

We can already see from Fig. 3 that three parameters have a larger magnitude, namely $\boldsymbol{\alpha}_3$, $\boldsymbol{\beta}_2$, and $\boldsymbol{\gamma}_3$. Interestingly, the mechanism performed by these three parameters together corresponds exactly to an induction head. In the first layer, $\boldsymbol{\alpha}_3$ makes each label attend to the preceding item. In the second layer, $\boldsymbol{\beta}_2$ makes the query item attend to the correct label based on the newly retrieved item. Finally, $\boldsymbol{\gamma}_3$ outputs the label retrieved by the second layer. In Fig. 4 (top), we visualize the 19 pseudo-parameters and loss during training, confirming that the drop in loss is driven by the emergence of the induction head.

**Hypothesis 2.** ***Self-Contained Dynamics.*** *The emergence of the induction head is unaided by the presence of any other parameter.*

By training the model while constraining its parameters to the 3-dimensional subspace spanned by the three parameters, we uncover very similar dynamics. As depicted in Fig. 4 (bottom), we find that the emergence of the induction head is unaffected, even slightly accelerated. We show a few more plots and full training details in App. D.

## 6 FULL TRAINING DYNAMICS OF INDUCTION HEADS

Motivated by the empirical results in the previous section, we study the training dynamics constrained to the 3-dimensional subspace spanned by $\boldsymbol{\alpha}_3$, $\boldsymbol{\beta}_2$, and $\boldsymbol{\gamma}_3$, finding several tight bounds for the emergence of the induction head.

### 6.1 THEORETICAL RESULTS

We study the emergence of an induction head under the following assumptions:

**Assumption 4.** ***Three-learnable Parameters.*** *Only parameters $\alpha_3$, $\beta_2$, and $\gamma_3$ are learnable. For the rest of the proof, we refer to these parameters as simply $\alpha$, $\beta$, and $\gamma$.*

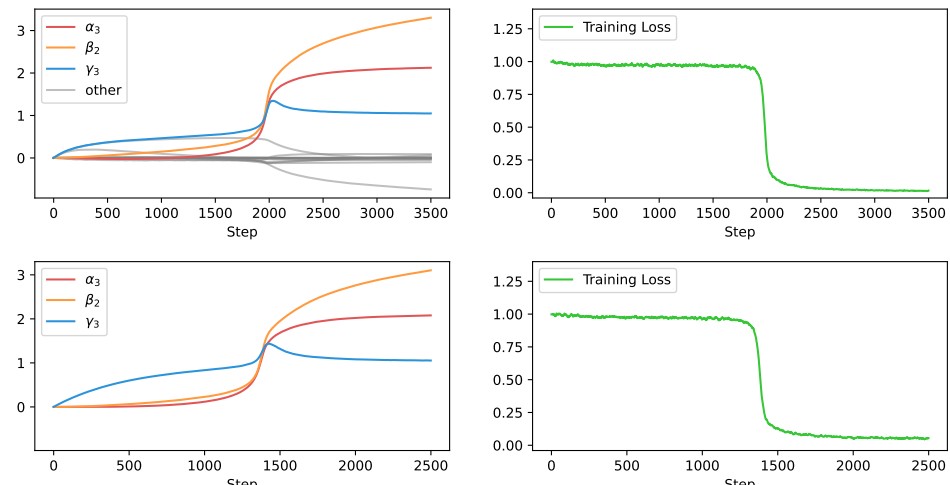

Figure 4: *Top:* The value of the 19 pseudo-parameters during standard training *(left)* and the associated training loss *(right)*. *Bottom:* Ablating all parameters except $\boldsymbol{\alpha}_3$, $\boldsymbol{\beta}_2$, and $\boldsymbol{\gamma}_3$ results in strikingly similar dynamics.

**Assumption 5. *Gradient Flow.*** *We study the training dynamics under the assumption of a continuous-time gradient flow with unit learning rate,*

$$\frac{\partial \alpha}{\partial t} = -\frac{\partial \mathcal{L}}{\partial \alpha}, \qquad \frac{\partial \beta}{\partial t} = -\frac{\partial \mathcal{L}}{\partial \beta}, \qquad \frac{\partial \gamma}{\partial t} = -\frac{\partial \mathcal{L}}{\partial \gamma},$$

*where $\alpha, \beta, \gamma : \mathbb{R}_{\geq 0} \to \mathbb{R}$ are the continuous-time trajectories of the three parameters.*

**Assumption 6. *Zero Initialization.*** *We assume our neural network is initialized with all weights having value zero. Equivalently, $\alpha(0) = \beta(0) = \gamma(0) = 0$.*

**Assumption 7. *Orthonormal Inputs.*** *We assume that all items, labels, and positional embeddings are orthogonal and have unit norm. Specifically,*

$$\|\boldsymbol{a}_i\| = \|\boldsymbol{b}_i\| = \|\boldsymbol{p}_i\| = 1, \quad \boldsymbol{a}_i^\mathsf{T} \boldsymbol{a}_j = \boldsymbol{b}_i^\mathsf{T} \boldsymbol{b}_j = \boldsymbol{a}_i^\mathsf{T} \boldsymbol{b}_i = \boldsymbol{p}_i^\mathsf{T} \boldsymbol{M} \boldsymbol{p}_i = \boldsymbol{p}_i^\mathsf{T} \boldsymbol{p}_j = \boldsymbol{p}_i^\mathsf{T} \boldsymbol{M} \boldsymbol{p}_j = 0,$$

*for all $i, j \in \{1, 2, \ldots, N\}, i \neq j$.*

Note that this assumption requires $D \geq 2N$. There are two ways to motivate this assumption, either by preprocessing the inputs using a whitening transformation, or by considering a very large dimension $D \to \infty$ and vectors sampled from an i.i.d. Gaussian with variance $1/\sqrt{D}$.

**Assumption 8. *Query Last.*** *We assume that the query item always refers to the last item-label pair present in the sequence, or $q = N$.*

Note that even if the target label's position is fixed, a full induction head is still required: the model cannot directly attend to specific positions because positional embeddings are randomly generated and carry no explicit location information.

**Definition 1. *Parameter Emergence Time.*** *We say that each of the parameters $\alpha$, $\beta$, or $\gamma$ has emerged when its value becomes greater than $1/2$ for the first time:*

$$T_\alpha = \inf\left\{t \;\middle|\; \alpha(t) \geq \tfrac{1}{2}\right\}, \quad T_\beta = \inf\left\{t \;\middle|\; \beta(t) \geq \tfrac{1}{2}\right\}, \quad T_\gamma = \inf\left\{t \;\middle|\; \gamma(t) \geq \tfrac{1}{2}\right\},$$

*where $t \in \mathbb{R}_{\geq 0}$.*

**Theorem 2.** *Assume that inputs are orthonormal and that only parameters $\alpha, \beta$, and $\gamma$ are learnable. In this case, we have that parameters always emerge in the order $T_\gamma < T_\beta < T_\alpha$ and the time until their emergence asymptotically follows:*

$$T_\alpha = \Theta\left(N^2\right), \qquad T_\beta = \Theta\left(N^2\right), \qquad T_\gamma = \Theta(N), \tag{10}$$

*where $N$ is the number of item-label pairs in the context.*

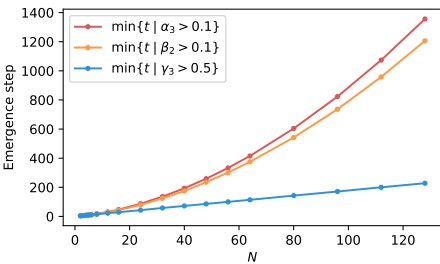 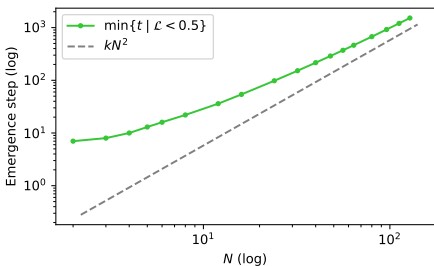

Figure 5: *Left:* The time until the emergence of $\boldsymbol{\alpha}_3$, $\boldsymbol{\beta}_2$, and $\boldsymbol{\gamma}_3$ for different values of $N$. *Right:* Time until the emergence of in-context learning (log scale) and its quadratic asymptote.

*Proof Sketch.* The proof is based on proving bounds for the gradient of each parameter. Before the emergence of any parameter, we have that $\partial\gamma/\partial t = \Theta(1/N)$, while $\partial\alpha/\partial t = O(1/N^2)$ and $\partial\beta/\partial t = O(1/N^2)$. This implies that $\gamma$ emerges first in $\Theta(N)$. Afterwards, we show that $\partial\beta/\partial t = \Theta(1/N^2)$ and $\partial\beta/\partial t > \partial\alpha/\partial t$. This implies that $\beta$ emerges next in $\Theta(N^2)$. Finally, we show that $\partial\alpha/\partial t = o(1/N^2)$, which implies that $\alpha$ emerges last in $\Theta(N^2)$. See the full proof in App. B. □

**Definition 2.** *Induction Head. We say that an induction has emerged if all three parameters are greater than* $1/2$.

**Definition 3.** *Time until ICL. We say that in-context learning has emerged at the first time when the induction head is present. Specifically,*

$$t_{\text{ICL}} = \inf\Big\{ t \in \mathbb{R}_{\geq 0} \ \Big| \ \alpha(t) \geq \tfrac{1}{2}, \ \beta(t) \geq \tfrac{1}{2}, \ \gamma(t) \geq \tfrac{1}{2} \Big\}.$$

**Corollary 1.** *The time until the emergence of in-context learning asymptotically follows:*

$$t_{ICL} = \Theta\Big(N^2\Big), \tag{11}$$

*where $N$ is the number of item-label pairs in the context.*

We empirically validate our theoretical results in Fig. 5. Training details in App. F.

## 7 DISCUSSION

### 7.1 HOW DO $\alpha$, $\beta$, AND $\gamma$ EMERGE DURING TRAINING?

**The emergence of $\gamma$.** Even if $\alpha$ and $\beta$ are completely untrained, the attention layers still return something: the average of all items and labels in the context. This average achieves a better loss than predicting zero, and this is exactly what the model learns to predict initially. However, this solution becomes worse when $N$ is increased. In fact, the gradient towards this solution is inversely proportional to $N$, hence why $\gamma$ emerges in $\Theta(N)$.

**The emergence of $\beta$.** After the final layer is in place, there is now a gradient for the second layer to attend correctly. Because each label follows immediately after its item, the first layer will always retrieve the item to some extent, even when completely untrained. Taking the causal masking into account, each item will be retrieved the most by its label. This enables the second layer to learn to retrieve based on the query item. However, since the first layer returns a very weak signal (inversely proportional to $N$), the gradient of $\beta$ will be inversely proportional to $N^2$.

**The emergence of $\alpha$.** Finally, after $\beta$ and $\gamma$ have emerged, there is a very strong gradient for the first layer to attend correctly. This quickly drives the emergence of $\alpha$.

### 7.2 THE IMPORTANCE OF CONTEXT LENGTH

We have established that a longer context length slows down the emergence of induction heads. This fact has interesting implications that are worth exploring in future work.

Chan et al. (2022) have empirically established that the emergence of in-context learning is modulated by data distributional properties specific to natural language, such as burstiness (items appear in clusters rather than being uniformly distributed over time). Our work paves the way for a theoretical understanding of this connection. For example, bustiness could be understood as a modulator of the *effective* context length by reducing the distance between items from the same class. We hypothesize that similar gains could be achieved by other means of reducing the *effective* context length, such as special positional embeddings (Su et al., 2024).

## 8 RELATED WORK

**In-Context Learning**  Brown et al. (2020) first observed that LLMs are capable of in-context learning. Since then, a number of works has delved deep into the phenomenon and its underlying causes. Chan et al. (2022) empirically showed that the ICL–IWL trade-off is modulated by data distributional properties specific to natural language, such as a Zipfian distribution over concepts, burstiness, and within-class variance. One direction is to view the forward pass of a transformer as performing gradient descent Von Oswald et al. (2023); Ahn et al. (2023). Finally, Lu et al. (2024) provides an asymptotic analysis of ICL for linear regression and linear attention.

**Induction Heads**  Later, Olsson et al. (2022) attributed this ability to a two-layer Sanford et al. (2024) mechanism (termed *induction head*) that emerges abruptly during training. Crucial to our work, Reddy (2023) proposed a 3-parameter *phenomenological* model of an induction head by directly parameterizing the attention scores. The parameters of this model (denoted as $\beta_1$, $\alpha$, and $\xi$) correspond exactly to our three pseudo-parameters ($\alpha_3$, $\beta_2$, and $\gamma_3$). Compared to their work, we provide a theoretical justification on how these parameters are learned with gradient descent. Other theoretical works have studied the emergence of induction heads, with different architectures and distributional assumptions Nichani et al. (2024a); Bietti et al. (2024); Chen et al. (2024); Sanford et al. (2024); Edelman et al. (2024); Wang et al. (2024a). Among these, Nichani et al. (2024a) demonstrates that two-layer disentangled transformers can learn to sample Markov chains in-context through a staged training process, and Bietti et al. (2024) study the transformer training dynamics from the perspective of *associative memories*. They show how an induction head can emerge after three steps of gradient descent. Concurrently, Chen et al. (2024) and Wang et al. (2024a) further studied staged layer-wise dynamics, reinforcing the staged learning hypothesis for induction head formation. Edelman et al. (2024) investigated how transformers acquire simple linguistic structures such as n-grams during training, and Zhang et al. (2025) analyzed training dynamics for linear attention transformers in regression tasks.

**Mechanistic Interpretability**  Mechanistic interpretability seeks to attribute the emergence of particular behaviors in neural networks to specific patterns in their weights and activations Olah et al. (2020); Elhage et al. (2021); Doshi-Velez & Kim (2017); Olah et al. (2017); Bereska & Gavves (2024); Cammarata et al. (2020). Friedman et al. (2023) introduce the *disentangled transformer* architecture, which is interpretable by design, but just as expressive. It keeps the residual stream disentangled by appending the attention output to the residual stream, rather than adding them together. Several works study transformers from the perspective of associative memories (Bietti et al., 2024; Nichani et al., 2024b; Chen et al., 2025). Other works focus on multi-step reasoning (Wang et al., 2024b; Mușat, 2025; Cabannes et al., 2024), context-free grammars (Allen-Zhu & Li, 2023), and modular addition (Nanda et al., 2023; Zhong et al., 2023; Gromov, 2023; He et al., 2024). Löwe et al. (2024) connect abrupt learning in artificial nets with insights in humans (also known as *evrika moments*).

## 9 CONCLUSION

In this paper, we have shown how induction heads emerge in an ICL task. Our work paves the way for a better theoretical understanding of transformer learning dynamics. We believe that a similar approach could illuminate other important phenomena in deep learning, such as the *in-context vs. in-weights learning* trade-off, abrupt learning, or the emergence of other transformer circuits.

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

## A    WEIGHTS STRUCTURE FULL PROOF

### A.1    SUMMARY

Our strategy is to show that if $W^{(1)}, W^{(2)}$, and $W^{(3)}$ have this structure, then their gradients also have the same structure. Since we start from zero initialization, by induction, this means that the structure is preserved throughout the entire training process.

To prove the structure of the gradient, we apply a carefully chosen rotation to the entire data distribution. Since the data distribution is isotropic, the rotation will not change the data distribution, so the expected gradient will also remain unchanged.

However, we are also able to show that our rotation induces a specific similarity transformation of the gradient:

$$\mathbb{E}\left[\frac{\partial \mathcal{L}}{\partial \boldsymbol{W}_{ij}^{(k)}}\right] \;=\; F\,\mathbb{E}\left[\frac{\partial \mathcal{L}}{\partial \boldsymbol{W}_{ij}^{(k)}}\right]\,F^{\top}$$

where $F$ is an orthogonal or block-orthogonal matrix and $\boldsymbol{W}_{ij}^{(k)}$ is a block of a weight matrix. From this we are able to show that the expected gradient must have the desired structure.

### A.2    PREREQUISITES

#### A.2.1    ORTHOGONAL TRANSFORMATIONS

**Definition 4.  *Orthogonal Matrix.*  *We say that a matrix $E \in \mathbb{R}^{k \times k}$ is orthogonal if it satisfies $EE^{\top} = E^{\top}E = I$.***

**Proposition 1.** *Let $A \in \mathbb{R}^{k \times k}$ be some matrix. If $EAE^{\top} = A$ holds **for all** orthogonal matrices $E \in \mathbb{R}^{k \times k}$, then it follows that $A = \alpha\,I$ for some $\alpha \in \mathbb{R}$.*

*Proof.  Step 1. All off-diagonal entries of $A$ vanish.*

Fix an index $j \in \{1, \ldots, k\}$ and let

$$E \;=\; \mathrm{diag}(1, \ldots, 1, -1, 1, \ldots, 1)$$

be the diagonal orthogonal matrix with entry $-1$ in the $j$th position and $+1$ elsewhere. Then

$$(EAE^{\top})_{i\ell} \;=\; E_{ii}\,A_{i\ell}\,E_{\ell\ell} \;=\; \begin{cases} A_{i\ell}, & i, \ell \neq j, \\ -A_{i\ell}, & \text{exactly one of } i, \ell = j, \\ A_{jj}, & i = \ell = j. \end{cases}$$

Since $EAE^{\top} = A$, it follows that $-A_{ij} = A_{ij}$ for every $i \neq j$, whence $A_{ij} = 0$. Varying $j$ shows all off-diagonal entries vanish, so

$$A = \mathrm{diag}(a_{11}, a_{22}, \ldots, a_{kk}).$$

*Step 2. All diagonal entries of $A$ coincide.*

Let $E$ be any permutation matrix which swaps two coordinates $i$ and $j$. Then $E$ is orthogonal and

$$EAE^{\top} = \mathrm{diag}(\ldots, a_{jj}, \ldots, a_{ii}, \ldots),$$

interchanging the $i$th and $j$th diagonal entries of $A$. By invariance $EAE^{\top} = A$, so $a_{ii} = a_{jj}$. Since $i, j$ were arbitrary, there exists $\alpha \in \mathbb{R}$ such that

$$a_{11} = a_{22} = \cdots = a_{kk} = \alpha,$$

and hence $A = \alpha I$.

$\square$

### A.2.2 BLOCK-ORTHOGONAL TRANSFORMATIONS

**Definition 5.** ***Block-Orthogonal Matrix.*** *We say that a matrix $F \in \mathbb{R}^{2k \times 2k}$ is block-orthogonal if it has either of the following two forms:*

$$F = \left[ \begin{array}{c|c} E & \mathbf{0} \\ \hline \mathbf{0} & E \end{array} \right] \qquad or \qquad F = \left[ \begin{array}{c|c} \mathbf{0} & E \\ \hline E & \mathbf{0} \end{array} \right]$$

*where $E \in \mathbb{R}^{k \times k}$ is an orthogonal matrix.*

**Proposition 2.** *Let $A \in \mathbb{R}^{2k \times 2k}$ be some matrix. If $FAF^\top = A$ holds **for all** block-orthogonal matrices $F \in \mathbb{R}^{2k \times 2k}$, then it follows that*

$$A = \left[ \begin{array}{c|c} \alpha I & \beta I \\ \hline \beta I & \alpha I \end{array} \right]$$

*for some $\alpha, \beta \in \mathbb{R}$.*

**Remark 1.** *Note that this condition is weaker than the condition stated in Proposition 1, since not all orthogonal matrices are also block-orthogonal. Hence, the condition in Proposition 2 guarantees a structure that is less specific than Proposition 1.*

*Proof.* We write

$$A = \left[ \begin{array}{c|c} A_{11} & A_{12} \\ \hline A_{21} & A_{22} \end{array} \right],$$

where each block $A_{ij} \in \mathbb{R}^{k \times k}$.

*Step 1. All blocks are scalar matrices.*

For any othogonal matrix $E$, we can set

$$F = \left[ \begin{array}{c|c} E & \mathbf{0} \\ \hline \mathbf{0} & E \end{array} \right].$$

Then

$$F A F^\top = \left[ \begin{array}{c|c} EA_{11}E^\top & EA_{12}E^\top \\ \hline EA_{21}E^\top & EA_{22}E^\top \end{array} \right] = \left[ \begin{array}{c|c} A_{11} & A_{12} \\ \hline A_{21} & A_{22} \end{array} \right] = A,$$

so $E A_{ij} E^\top = A_{ij}$ for all $i, j$. By the previous proposition each block is a scalar multiple of the identity, $A_{ij} = \alpha_{ij} I_k$, for some $\alpha_{ij} \in \mathbb{R}$. Therefore,

$$A = \left[ \begin{array}{c|c} \alpha_{11} I & \alpha_{12} I \\ \hline \alpha_{21} I & \alpha_{22} I \end{array} \right].$$

*Step 2. Diagonally opposed blocks coincide.* By setting

$$F = \left[ \begin{array}{c|c} \mathbf{0} & I \\ \hline I & \mathbf{0} \end{array} \right]$$

we obtain

$$F A F^\top = \left[ \begin{array}{c|c} A_{22} & A_{21} \\ \hline A_{12} & A_{11} \end{array} \right]$$

which yields $\alpha_{11} = \alpha_{22}$, $\alpha_{12} = \alpha_{21}$. By writing $\alpha = \alpha_{11}$ and $\beta = \alpha_{12}$, we obtain

$$A = \left[ \begin{array}{c|c} \alpha I & \beta I \\ \hline \beta I & \alpha I \end{array} \right]$$

$\square$

### A.2.3 COMBINED TRANSFORMATIONS

**Proposition 3.** *Let $A \in \mathbb{R}^{2k \times 2k}$ be some matrix. If $EAF = A$ holds for all orthogonal matrices $E$ and block-orthogonal matrices $F$, then $A = \mathbf{0}$.*

*Proof.* By setting $E = I$ and $F = -I$, we get $A = -A$. Therefore, $A = \mathbf{0}$. $\square$

### A.2.4 BLOCK-SWAP TRANSFORMATION

**Definition 6.** ***Block-Swap Matrix.*** *We say that a matrix $M \in \mathbb{R}^{2k \times 2k}$ is block-swap if it has the following form:*

$$M = \left[ \begin{array}{c|c} \mathbf{0} & I \\ \hline I & \mathbf{0} \end{array} \right]$$

*where $I \in \mathbb{R}^{k \times k}$ is the identity matrix.*

**Proposition 4.** *If $M \in \mathbb{R}^{2k \times 2k}$ is a block-swap matrix and $F \in \mathbb{R}^{2k \times 2k}$ is a block-orthogonal matrix, then $FMF^\mathsf{T} = M$.*

*Proof. Case 1. The orthogonal blocks of F are on the main diagonal.*

Assume that

$$F = \left[ \begin{array}{c|c} E & \mathbf{0} \\ \hline \mathbf{0} & E \end{array} \right]$$

Then,

$$
\begin{aligned}
FMF^\mathsf{T} &= \left[ \begin{array}{c|c} E & \mathbf{0} \\ \hline \mathbf{0} & E \end{array} \right] \left[ \begin{array}{c|c} \mathbf{0} & I \\ \hline I & \mathbf{0} \end{array} \right] \left[ \begin{array}{c|c} E^\mathsf{T} & \mathbf{0} \\ \hline \mathbf{0} & E^\mathsf{T} \end{array} \right] \\
&= \left[ \begin{array}{c|c} \mathbf{0} & E \\ \hline E & \mathbf{0} \end{array} \right] \left[ \begin{array}{c|c} E^\mathsf{T} & \mathbf{0} \\ \hline \mathbf{0} & E^\mathsf{T} \end{array} \right] \\
&= \left[ \begin{array}{c|c} \mathbf{0} & I \\ \hline I & \mathbf{0} \end{array} \right]
\end{aligned}
$$

*Case 2. The orthogonal blocks of F are on the secondary diagonal.*

Assume that

$$F = \left[ \begin{array}{c|c} \mathbf{0} & E \\ \hline E & \mathbf{0} \end{array} \right]$$

Then,

$$
\begin{aligned}
FMF^\mathsf{T} &= \left[ \begin{array}{c|c} \mathbf{0} & E \\ \hline E & \mathbf{0} \end{array} \right] \left[ \begin{array}{c|c} \mathbf{0} & I \\ \hline I & \mathbf{0} \end{array} \right] \left[ \begin{array}{c|c} \mathbf{0} & E^\mathsf{T} \\ \hline E^\mathsf{T} & \mathbf{0} \end{array} \right] \\
&= \left[ \begin{array}{c|c} E & \mathbf{0} \\ \hline \mathbf{0} & E \end{array} \right] \left[ \begin{array}{c|c} \mathbf{0} & E^\mathsf{T} \\ \hline E^\mathsf{T} & \mathbf{0} \end{array} \right] \\
&= \left[ \begin{array}{c|c} \mathbf{0} & I \\ \hline I & \mathbf{0} \end{array} \right]
\end{aligned}
$$

$\square$

### A.3 SETUP

Recall the architecture and loss:

$$U = \left[ \; X \; \middle| \; \sigma(XW^{(1)}X^\mathsf{T})\,X \; \right] \qquad\qquad V = \left[ \; U \; \middle| \; \sigma(UW^{(2)}U^\mathsf{T})\,U \; \right]$$

$$z = V_{2N+1}W^{(3)} \qquad \mathcal{L} = \|y - z\|^2$$

where $\sigma$ to denotes the softmax function with causal masking, $[\,\cdot\,|\,\cdot\,]$ denotes matrix concatenation, and

$$W^{(1)} \in \mathbb{R}^{2D \times 2D} \qquad\qquad W^{(2)} \in \mathbb{R}^{4D \times 4D} \qquad\qquad W^{(3)} \in \mathbb{R}^{8D \times D}$$

$$U \in \mathbb{R}^{(2N+1) \times 4D} \qquad\qquad V \in \mathbb{R}^{(2N+1) \times 8D} \qquad\qquad z \in \mathbb{R}^D$$

$$X \in \mathbb{R}^{(2N+1) \times 2D} \qquad\qquad y \in \mathbb{R}^D$$

The data is generated as:

$$X_{2i-1} = \begin{bmatrix} a_i \mid p_i \end{bmatrix} \qquad X_{2i} = \begin{bmatrix} b_i \mid p_i M \end{bmatrix} \qquad \forall i \in \{1, \ldots, N\}$$

$$X_{2N+1} = \begin{bmatrix} a_q \mid 0 \end{bmatrix} \qquad y = b_q$$

where

$$a_i,\ b_i,\ p_i\ \in\ \mathbb{R}^D \qquad q \in \{1, 2, \ldots, N\} \qquad M = \begin{bmatrix} \mathbf{0} & I \\ I & \mathbf{0} \end{bmatrix}$$

All vectors are treated as **row vectors**.

## A.4 Additional Notation

We introduce

$$\begin{aligned} S &= X\boldsymbol{W}^{(1)}X^\intercal & T &= \sigma(S) \\ P &= U\boldsymbol{W}^{(2)}U^\intercal & Q &= \sigma(P) \end{aligned}$$

where $S,\ T,\ P,\ Q\ \in\ \mathbb{R}^{(2N+1)\times(2N+1)}$. This gives

$$U = \begin{bmatrix} X \mid TX \end{bmatrix} \qquad\qquad V = \begin{bmatrix} U \mid QU \end{bmatrix}$$

We also introduce notation for all blocks of size $D$:

$$X = \begin{bmatrix} X_1 & X_2 \end{bmatrix} \qquad U = \begin{bmatrix} U_1 & U_2 & U_3 & U_4 \end{bmatrix}$$

$$V = \begin{bmatrix} V_1 & V_2 & V_3 & V_4 & V_5 & V_6 & V_7 & V_8 \end{bmatrix}$$

$$\boldsymbol{W}^{(1)} = \begin{bmatrix} \boldsymbol{W}^{(1)}_{11} & \boldsymbol{W}^{(1)}_{12} \\ \boldsymbol{W}^{(1)}_{21} & \boldsymbol{W}^{(1)}_{22} \end{bmatrix} \qquad \boldsymbol{W}^{(2)} = \begin{bmatrix} \boldsymbol{W}^{(2)}_{11} & \boldsymbol{W}^{(2)}_{12} & \boldsymbol{W}^{(2)}_{13} & \boldsymbol{W}^{(2)}_{14} \\ \boldsymbol{W}^{(2)}_{11} & \boldsymbol{W}^{(2)}_{12} & \boldsymbol{W}^{(2)}_{13} & \boldsymbol{W}^{(2)}_{14} \\ \boldsymbol{W}^{(2)}_{11} & \boldsymbol{W}^{(2)}_{12} & \boldsymbol{W}^{(2)}_{13} & \boldsymbol{W}^{(2)}_{14} \\ \boldsymbol{W}^{(2)}_{11} & \boldsymbol{W}^{(2)}_{12} & \boldsymbol{W}^{(2)}_{13} & \boldsymbol{W}^{(2)}_{14} \end{bmatrix}$$

$$\boldsymbol{W}^{(3)} = \begin{bmatrix} \boldsymbol{W}^{(3)}_1 & \boldsymbol{W}^{(3)}_2 & \boldsymbol{W}^{(3)}_3 & \boldsymbol{W}^{(3)}_4 & \boldsymbol{W}^{(3)}_5 & \boldsymbol{W}^{(3)}_6 & \boldsymbol{W}^{(3)}_7 & \boldsymbol{W}^{(3)}_8 \end{bmatrix}$$

## A.5 Data Rotations

We apply an orthogonal transformation $E$ to the items and labels, and a block-orthogonal transformation $F$ to the positional embeddings:

$$a_i' = a_i E \qquad b_i' = b_i E \qquad p_i' = p_i F \qquad \forall i \in \{\,1,\ \ldots,\ N\,\}$$

where $E$ and $F$ satisfy Definitions 4 and 5, respectively. We refer to the new variables as $X'$, $y'$, $U'$, $V'$, $z'$, and $\mathcal{L}'$.

Since the data is isotropic, we have that $\mathbb{E}[\mathcal{L}] = \mathbb{E}[\mathcal{L}']$. By the linearity of expectation and differentiation, we obtain

$$\mathbb{E}\left[\frac{\partial \mathcal{L}}{\partial \boldsymbol{W}^{(k)}}\right] = \mathbb{E}\left[\frac{\partial \mathcal{L}'}{\partial \boldsymbol{W}^{(k)}}\right]$$

This also holds for all sub-blocks of $\boldsymbol{W}^{(1)}$, $\boldsymbol{W}^{(2)}$, and $\boldsymbol{W}^{(3)}$,

$$\mathbb{E}\left[\frac{\partial \mathcal{L}}{\partial \boldsymbol{W}_{ij}^{(k)}}\right] \;=\; \mathbb{E}\left[\frac{\partial \mathcal{L}'}{\partial \boldsymbol{W}_{ij}^{(k)}}\right]$$

However, as we show below, our rotation induces specific transformations of the gradient blocks. Using Propositions 1 to 3, we are able to show that each gradient block has the desired structure.

Specifically, for each gradient block, we will show that one of the following four conditions holds for all $E$ and $F$, implying the desired structure:

$$\mathbb{E}\left[\frac{\partial \mathcal{L}'}{\partial \boldsymbol{W}_{ij}^{(k)}}\right] \;=\; E\,\mathbb{E}\left[\frac{\partial \mathcal{L}'}{\partial \boldsymbol{W}_{ij}^{(k)}}\right]E^{\intercal} \quad \implies \quad \mathbb{E}\left[\frac{\partial \mathcal{L}}{\partial \boldsymbol{W}_{ij}^{(k)}}\right] \;=\; \alpha\,I$$

$$\mathbb{E}\left[\frac{\partial \mathcal{L}'}{\partial \boldsymbol{W}_{ij}^{(k)}}\right] \;=\; F\,\mathbb{E}\left[\frac{\partial \mathcal{L}'}{\partial \boldsymbol{W}_{ij}^{(k)}}\right]F^{\intercal} \quad \implies \quad \mathbb{E}\left[\frac{\partial \mathcal{L}}{\partial \boldsymbol{W}_{ij}^{(k)}}\right] \;=\; \left[\begin{array}{cc} \alpha\,I & \beta\,I \\ \beta\,I & \alpha\,I \end{array}\right]$$

$$\mathbb{E}\left[\frac{\partial \mathcal{L}'}{\partial \boldsymbol{W}_{ij}^{(k)}}\right] \;=\; E\,\mathbb{E}\left[\frac{\partial \mathcal{L}'}{\partial \boldsymbol{W}_{ij}^{(k)}}\right]F^{\intercal} \quad \implies \quad \mathbb{E}\left[\frac{\partial \mathcal{L}}{\partial \boldsymbol{W}_{ij}^{(k)}}\right] \;=\; \boldsymbol{0}$$

$$\mathbb{E}\left[\frac{\partial \mathcal{L}'}{\partial \boldsymbol{W}_{ij}^{(k)}}\right] \;=\; F\,\mathbb{E}\left[\frac{\partial \mathcal{L}'}{\partial \boldsymbol{W}_{ij}^{(k)}}\right]E^{\intercal} \quad \implies \quad \mathbb{E}\left[\frac{\partial \mathcal{L}}{\partial \boldsymbol{W}_{ij}^{(k)}}\right] \;=\; \boldsymbol{0}$$

## A.6 FORWARD PASS

We will now observe how our rotation changes the intermediate and final results of our model.

First, note the rotated inputs and outputs:

$$X_1' = X_1 E \qquad X_2' = X_2 F \qquad y' = yE$$

Recall that we are assuming that $\boldsymbol{W}^{(1)}$, $\boldsymbol{W}^{(2)}$, and $\boldsymbol{W}^{(3)}$ already have the desired structure, with the goal to prove that the gradient has the same structure:

$$W^{(1)} \;=\; \left[\begin{array}{c|c} \alpha_1 I & \boldsymbol{0} \\ \hline \boldsymbol{0} & \alpha_2 I + \alpha_3 M \end{array}\right]$$

$$W^{(2)} \;=\; \left[\begin{array}{c|c|c|c} \beta_1 I & \boldsymbol{0} & \beta_2 I & \boldsymbol{0} \\ \hline \boldsymbol{0} & \beta_3 I + \beta_4 M & \boldsymbol{0} & \beta_5 I + \beta_6 M \\ \hline \beta_7 I & \boldsymbol{0} & \beta_8 I & \boldsymbol{0} \\ \hline \boldsymbol{0} & \beta_9 I + \beta_{10} M & \boldsymbol{0} & \beta_{11} I + \beta_{12} M \end{array}\right]$$

$$W^{(3)} \;=\; \left[\; \gamma_1 I \mid \boldsymbol{0} \mid \gamma_2 I \mid \boldsymbol{0} \mid \gamma_3 I \mid \boldsymbol{0} \mid \gamma_4 I \mid \boldsymbol{0} \;\right]$$

### A.6.1 FIRST LAYER

The first attention layer gives:

$$\begin{aligned}
S \;=\;& X \boldsymbol{W}^{(1)} X^{\intercal} \\
=\;& X_1 \boldsymbol{W}_{11}^{(1)} X_1^{\intercal} + X_1 \boldsymbol{W}_{12}^{(1)} X_2^{\intercal} + X_2 \boldsymbol{W}_{21}^{(1)} X_1^{\intercal} + X_2 \boldsymbol{W}_{22}^{(1)} X_2^{\intercal} \\
=\;& \alpha_1 X_1 X_1^{\intercal} + \alpha_2 X_2 X_2^{\intercal} + \alpha_3 X_2 M X_2^{\intercal} \\[4pt]
S' \;=\;& X' \boldsymbol{W}^{(1)} X'^{\intercal} \\
=\;& X_1' \boldsymbol{W}_{11}^{(1)} X_1'^{\intercal} + X_1' \boldsymbol{W}_{12}^{(1)} X_2'^{\intercal} + X_2' \boldsymbol{W}_{21}^{(1)} X_1'^{\intercal} + X_2' \boldsymbol{W}_{22}^{(1)} X_2'^{\intercal} \\
=\;& \alpha_1 X_1' X_1'^{\intercal} + \alpha_2 X_2' X_2'^{\intercal} + \alpha_3 X_2' M X_2'^{\intercal} \\
=\;& \alpha_1 X_1 E E^{\intercal} X_1^{\intercal} + \alpha_2 X_2 F F^{\intercal} X_2^{\intercal} + \alpha_3 X_2 F M F^{\intercal} X_2^{\intercal} \\
=\;& \alpha_1 X_1 X_1^{\intercal} + \alpha_2 X_2 X_2^{\intercal} + \alpha_3 X_2 M X_2^{\intercal}
\end{aligned}$$

Therefore, $S' = S$ and $T' = T = \sigma(S)$. This gives us:

$$U_1' = U_1 E \qquad U_2' = U_2 F \qquad U_3' = U_3 E \qquad U_4' = U_4 F$$

### A.6.2 SECOND LAYER

The second attention layer gives:

$$
\begin{aligned}
P &= U \boldsymbol{W}^{(2)} U^\intercal \\
&= \sum U_i \boldsymbol{W}^{(2)}_{ij} U_j{}^\intercal \\
&= \beta_1 U_1 U_1{}^\intercal + \beta_2 U_1 U_3^\intercal + \beta_7 U_3 U_1^\intercal + \beta_8 U_3 U_3^\intercal \\
&\quad + \beta_3 U_2 U_2^\intercal + \beta_5 U_2 U_4^\intercal + \beta_9 U_4 U_2^\intercal + \beta_{11} U_4 U_4^\intercal \\
&\quad + \beta_4 U_2 M U_2^\intercal + \beta_6 U_2 M U_4^\intercal + \beta_{10} U_4 M U_2^\intercal + \beta_{12} U_4 M U_4^\intercal
\end{aligned}
$$

$$
\begin{aligned}
P' &= U' \boldsymbol{W}^{(2)} U'^\intercal \\
&= \sum U_i' \boldsymbol{W}^{(2)}_{ij} U_j'^\intercal \\
&= \beta_1 U_1' U_1'^\intercal + \beta_2 U_1' U_3'^\intercal + \beta_7 U_3' U_1'^\intercal + \beta_8 U_3' U_3'^\intercal \\
&\quad + \beta_3 U_2' U_2'^\intercal + \beta_5 U_2' U_4'^\intercal + \beta_9 U_4' U_2'^\intercal + \beta_{11} U_4' U_4'^\intercal \\
&\quad + \beta_4 U_2' M U_2'^\intercal + \beta_6 U_2' M U_4'^\intercal + \beta_{10} U_4' M U_2'^\intercal + \beta_{12} U_4' M U_4'^\intercal \\
&= \beta_1 U_1 E E^\intercal U_1^\intercal + \beta_2 U_1 E E^\intercal U_3^\intercal + \beta_7 U_3 E E^\intercal U_1^\intercal + \beta_8 U_3 E E^\intercal U_3^\intercal \\
&\quad + \beta_3 U_2 F F^\intercal U_2^\intercal + \beta_5 U_2 F F^\intercal U_4^\intercal + \beta_9 U_4 F F^\intercal U_2^\intercal + \beta_{11} U_4 F F^\intercal U_4^\intercal \\
&\quad + \beta_4 U_2 F M F^\intercal U_2^\intercal + \beta_6 U_2 F M F^\intercal U_4^\intercal + \beta_{10} U_4 F M F^\intercal U_2^\intercal + \beta_{12} F M F^\intercal U_4^\intercal \\
&= \beta_1 U_1 U_1{}^\intercal + \beta_2 U_1 U_3^\intercal + \beta_7 U_3 U_1^\intercal + \beta_8 U_3 U_3^\intercal \\
&\quad + \beta_3 U_2 U_2^\intercal + \beta_5 U_2 U_4^\intercal + \beta_9 U_4 U_2^\intercal + \beta_{11} U_4 U_4^\intercal \\
&\quad + \beta_4 U_2 M U_2^\intercal + \beta_6 U_2 M U_4^\intercal + \beta_{10} U_4 M U_2^\intercal + \beta_{12} U_4 M U_4^\intercal
\end{aligned}
$$

Therefore, $P' = P$ and $Q' = Q = \sigma(P)$. This gives us:

$$V_1' = V_1 E \qquad V_2' = V_2 F \qquad V_3' = V_3 E \qquad V_4' = V_4 F$$

$$V_5' = V_5 E \qquad V_6' = V_6 F \qquad V_7' = V_7 E \qquad V_8' = V_8 F$$

### A.6.3 OUTPUT LAYER

Finally, the output layer gives:

$$
\begin{aligned}
z &= V_{2N+1} \boldsymbol{W}^{(3)} \\
&= \sum (V_i)_{2N+1} \boldsymbol{W}^{(3)}_i \\
&= \gamma_1 (V_1)_{2N+1} + \gamma_2 (V_3)_{2N+1} + \gamma_3 (V_5)_{2N+1} + \gamma_4 (V_7)_{2N+1}
\end{aligned}
$$

$$
\begin{aligned}
z' &= V_{2N+1}' \boldsymbol{W}^{(3)} \\
&= \sum (V_i')_{2N+1} \boldsymbol{W}^{(3)}_i \\
&= \gamma_1 (V_1')_{2N+1} + \gamma_2 (V_3')_{2N+1} + \gamma_3 (V_5')_{2N+1} + \gamma_4 (V_7')_{2N+1} \\
&= \gamma_1 (V_1)_{2N+1} E + \gamma_2 (V_3)_{2N+1} E + \gamma_3 (V_5)_{2N+1} E + \gamma_4 (V_7)_{2N+1} E \\
&= z E
\end{aligned}
$$

### A.7 BACKWARD PASS

We now show how the rotation transforms the gradient of each weight block.

### A.7.1 OUTPUT LAYER

$$\frac{\partial \mathcal{L}}{\partial z} = 2(z - y)$$

$$\frac{\partial \mathcal{L}'}{\partial z'} = 2(z' - y') = 2(zE - yE) = 2(z - y)E = \frac{\partial \mathcal{L}}{\partial z}E$$

$$\frac{\partial \mathcal{L}}{\partial \boldsymbol{W}_i^{(3)}} = ((V_i)_{2N+1})^\intercal \left(\frac{\partial \mathcal{L}}{\partial z}\right)$$

$$\frac{\partial \mathcal{L}'}{\partial \boldsymbol{W}_i^{(3)}} = ((V_i')_{2N+1})^\intercal \left(\frac{\partial \mathcal{L}'}{\partial z'}\right)$$

**Scalar Blocks.** For all $i \in \{1, 3, 5, 7\}$, we get

$$\frac{\partial \mathcal{L}'}{\partial \boldsymbol{W}_i^{(3)}} = ((V_i')_{2N+1})^\intercal \left(\frac{\partial \mathcal{L}'}{\partial z'}\right)$$

$$= E^\intercal ((V_i)_{2N+1})^\intercal \left(\frac{\partial \mathcal{L}}{\partial z}\right) E$$

$$= E^\intercal \frac{\partial \mathcal{L}}{\partial \boldsymbol{W}_i^{(3)}} E$$

Taking the expectation over the entire data distribution, we obtain that the following holds for any orthogonal transformation $E$:

$$\mathbb{E}\left[\frac{\partial \mathcal{L}}{\partial \boldsymbol{W}_i^{(3)}}\right] = \mathbb{E}\left[\frac{\partial \mathcal{L}'}{\partial \boldsymbol{W}_i^{(3)}}\right] = \mathbb{E}\left[E^\intercal \frac{\partial \mathcal{L}'}{\partial \boldsymbol{W}_i^{(3)}} E\right] = E^\intercal \mathbb{E}\left[\frac{\partial \mathcal{L}}{\partial \boldsymbol{W}_i^{(3)}}\right] E$$

Applying Proposition 1, we get that

$$\mathbb{E}\left[\frac{\partial \mathcal{L}}{\partial \boldsymbol{W}_i^{(3)}}\right] = \alpha \, I$$

**Zero Blocks.** For all $i \in \{2, 4, 6, 8\}$, we get

$$\frac{\partial \mathcal{L}'}{\partial \boldsymbol{W}_i^{(3)}} = ((V_i')_{2N+1})^\intercal \left(\frac{\partial \mathcal{L}'}{\partial z'}\right)$$

$$= F^\intercal ((V_i)_{2N+1})^\intercal \left(\frac{\partial \mathcal{L}}{\partial z}\right) E$$

$$= F^\intercal \frac{\partial \mathcal{L}}{\partial \boldsymbol{W}_i^{(3)}} E$$

Taking the expectation over the entire data distribution, we obtain that the following holds for any $E$ and $F$:

$$\mathbb{E}\left[\frac{\partial \mathcal{L}}{\partial \boldsymbol{W}_i^{(3)}}\right] = F^\intercal \mathbb{E}\left[\frac{\partial \mathcal{L}}{\partial \boldsymbol{W}_i^{(3)}}\right] E$$

Applying Proposition 3, we get that

$$\mathbb{E}\left[\frac{\partial \mathcal{L}}{\partial \boldsymbol{W}_i^{(3)}}\right] \;=\; \boldsymbol{0}$$

**Gradient Propagation** Applying the chain rule, we get

$$\frac{\partial \mathcal{L}}{\partial V_{2N+1}} \;=\; \frac{\partial \mathcal{L}}{\partial z}\frac{\partial z}{\partial V_{2N+1}} \;=\; 2(z-y)\,\boldsymbol{W}^{(3)\mathsf{T}}$$

For all $i \in \{1,3,5,7\}$, we get

$$\frac{\partial \mathcal{L}'}{\partial (V_i')_{2N+1}} \;=\; 2(z'-y')\,\boldsymbol{W}_i^{(3)\mathsf{T}} \;=\; 2(z-y)\,E\,\boldsymbol{W}_i^{(3)\mathsf{T}} \;=\; \frac{\partial \mathcal{L}}{\partial (V_i)_{2N+1}}E$$

For all $i \in \{2,4,6,8\}$, we get

$$\frac{\partial \mathcal{L}}{\partial (V_i)_{2N+1}} \;=\; 2(z-y)\,\boldsymbol{W}_i^{(3)} \;=\; \boldsymbol{0}$$

For all $i \in \{1,\dots,8\}$ and $j \le 2N$, we get

$$\frac{\partial \mathcal{L}}{\partial (V_i)_j} \;=\; \frac{\partial \mathcal{L}}{\partial (V_i')_j} \;=\; \boldsymbol{0}$$

Putting everything together, the following holds for all $j \le 2N+1$

$$\frac{\partial \mathcal{L}}{\partial (V_i)_j} \;=\; \frac{\partial \mathcal{L}'}{\partial (V_i')_j}E \qquad\qquad \text{if } i \in \{1,3,5,7\} \tag{12}$$

$$\frac{\partial \mathcal{L}}{\partial (V_i)_j} \;=\; \frac{\partial \mathcal{L}'}{\partial (V_i')_j} \;=\; \boldsymbol{0} \qquad\qquad \text{if } i \in \{2,4,6,8\} \tag{13}$$

### A.7.2 SECOND LAYER

Since $V = [\,U \mid QU\,]$, we have that

$$\frac{\partial (V_i)_j}{\partial Q_{jk}} \;=\; \begin{cases} U_{i-4} & i > 4 \\ \boldsymbol{0} & i \le 4 \end{cases}$$

Therefore,

$$\frac{\partial \mathcal{L}'}{\partial (V_i')_j}\frac{\partial (V_i')_j}{\partial Q_{jk}'} \;=\; \frac{\partial \mathcal{L}}{\partial (V_i)_j}E^{\mathsf{T}}E\,\frac{\partial (V_i)_j}{\partial Q_{jk}} \;=\; \frac{\partial \mathcal{L}}{\partial (V_i)_j}\frac{\partial (V_i)_j}{\partial Q_{jk}} \qquad i \in \{5,7\}$$

$$\frac{\partial \mathcal{L}'}{\partial (V_i')_j}\frac{\partial (V_i')_j}{\partial Q_{jk}'} \;=\; \frac{\partial \mathcal{L}}{\partial (V_i)_j}\frac{\partial (V_i)_j}{\partial Q_{jk}} \;=\; \boldsymbol{0} \qquad i \notin \{5,7\}$$

Additionally, since $P' = P$ and $Q' = Q$, we have that

$$\frac{\partial Q_{jk}'}{\partial P_{jl}'} = \frac{\partial Q_{jk}}{\partial P_{jl}}$$

This gives us

$$\frac{\partial \mathcal{L}'}{\partial P_{kl}'} \;=\; \sum_{ij}\frac{\partial \mathcal{L}'}{\partial (V_i')_j}\frac{\partial (V_i')_j}{\partial Q_{kj}'}\frac{\partial Q_{kj}'}{\partial P_{kl}'}$$

$$=\; \sum_{ij}\frac{\partial \mathcal{L}}{\partial (V_i)_k}\frac{\partial (V_i)_k}{\partial Q_{kj}}\frac{\partial Q_{kj}}{\partial P_{kl}}$$

$$=\; \frac{\partial \mathcal{L}}{\partial P_{kl}}$$

Additionally,

$$\frac{\partial \mathcal{L}}{\partial \boldsymbol{W}_{ij}^{(2)}} \;=\; \sum_{kl} \frac{\partial \mathcal{L}}{\partial P_{kl}} \frac{\partial P_{kl}}{\partial \boldsymbol{W}_{ij}^{(2)}} \;=\; \sum_{kl} \frac{\partial \mathcal{L}}{\partial P_{kl}} (U_i)_k^{\mathsf{T}} (U_j)_l$$

which gives us the transformed gradient:

$$\frac{\partial \mathcal{L}'}{\partial \boldsymbol{W}_{ij}^{(2)}} \;=\; \sum_{kl} \frac{\partial \mathcal{L}'}{\partial P_{kl}'} \frac{\partial P_{kl}'}{\partial \boldsymbol{W}_{ij}^{(2)}} \;=\; \sum_{kl} \frac{\partial \mathcal{L}}{\partial P_{kl}} (U_i')_k^{\mathsf{T}} (U_j')_l$$

$$= \begin{cases} E^{\mathsf{T}} \dfrac{\partial \mathcal{L}}{\partial \boldsymbol{W}_{ij}^{(2)}} E & \text{if } i \text{ odd}, j \text{ odd} \\[2.2ex] E^{\mathsf{T}} \dfrac{\partial \mathcal{L}}{\partial \boldsymbol{W}_{ij}^{(2)}} F & \text{if } i \text{ odd}, j \text{ even} \\[2.2ex] F^{\mathsf{T}} \dfrac{\partial \mathcal{L}}{\partial \boldsymbol{W}_{ij}^{(2)}} E & \text{if } i \text{ even}, j \text{ odd} \\[2.2ex] F^{\mathsf{T}} \dfrac{\partial \mathcal{L}}{\partial \boldsymbol{W}_{ij}^{(2)}} F & \text{if } i \text{ even}, j \text{ even} \end{cases}$$

The desired structure follows from computing the expected gradient over the entire distribution and applying Propositions 1 to 3.

**Gradient Propagation**

Applying the chain rule, we get

$$\frac{\partial \mathcal{L}'}{\partial (U_i')_j} \;=\; \frac{\partial \mathcal{L}'}{\partial (V_i')_j} + Q'^{\mathsf{T}} \frac{\partial \mathcal{L}'}{\partial (V_{i+4}')_j} + \sum_k \frac{\partial \mathcal{L}'}{\partial P_{jk}'} \frac{\partial P_{jk}'}{\partial (U_i')_j} \tag{14}$$

We also have that

$$\frac{\partial P_{jk}}{\partial (U_i)_j} \;=\; (U_i)_k \qquad\qquad \frac{\partial P_{jk}'}{\partial (U_i')_j} \;=\; (U_i')_k \;=\; \begin{cases} (U_i)_k \, E & \text{if } i \text{ odd} \\[1.5ex] (U_i)_k \, F & \text{if } i \text{ even} \end{cases} \tag{15}$$

Cobmining eqs. (12) to (15), we get

$$\frac{\partial \mathcal{L}'}{\partial (U_i')_j} \;=\; \begin{cases} \dfrac{\partial \mathcal{L}'}{\partial (U_i')_j} E & \text{if } i \text{ odd} \\[2.2ex] \dfrac{\partial \mathcal{L}'}{\partial (U_i')_j} F & \text{if } i \text{ even} \end{cases}$$

### A.7.3 FIRST LAYER

Through similar derivations as before, we obtain

$$\frac{\partial \mathcal{L}'}{\partial S_{kl}'} \;=\; \sum_{ij} \frac{\partial \mathcal{L}'}{\partial (U_i')_j} \frac{\partial (U_i')_j}{\partial T_{kj}'} \frac{\partial T_{kj}'}{\partial S_{kl}'}$$

$$=\; \sum_{ij} \frac{\partial \mathcal{L}}{\partial (U_i)_k} \frac{\partial (U_i)_k}{\partial T_{kj}} \frac{\partial T_{kj}}{\partial S_{kl}}$$

$$=\; \frac{\partial \mathcal{L}}{\partial S_{kl}}$$

and

$$\frac{\partial \mathcal{L}}{\partial \boldsymbol{W}_{ij}^{(1)}} = \sum_{kl} \frac{\partial \mathcal{L}}{\partial S_{kl}} \frac{\partial S_{kl}}{\partial \boldsymbol{W}_{ij}^{(1)}} = \sum_{kl} \frac{\partial \mathcal{L}}{\partial S_{kl}} (X_i)_k^{\mathsf{T}} (X_j)_l$$

This gives us the transformed gradient:

$$\frac{\partial \mathcal{L}'}{\partial \boldsymbol{W}_{11}^{(1)}} = E^{\mathsf{T}} \frac{\partial \mathcal{L}}{\partial \boldsymbol{W}_{11}^{(1)}} E \qquad\qquad \frac{\partial \mathcal{L}'}{\partial \boldsymbol{W}_{12}^{(1)}} = E^{\mathsf{T}} \frac{\partial \mathcal{L}}{\partial \boldsymbol{W}_{12}^{(1)}} F$$

$$\frac{\partial \mathcal{L}'}{\partial \boldsymbol{W}_{21}^{(1)}} = F^{\mathsf{T}} \frac{\partial \mathcal{L}}{\partial \boldsymbol{W}_{21}^{(1)}} E \qquad\qquad \frac{\partial \mathcal{L}'}{\partial \boldsymbol{W}_{22}^{(1)}} = F^{\mathsf{T}} \frac{\partial \mathcal{L}}{\partial \boldsymbol{W}_{22}^{(1)}} F$$

which implies the desired structure.

# B  TIGHT BOUND PROOF

## B.1  SUMMARY

We show that $\gamma$ is the first parameter to reach the value $1/2$ after a time $T_1 = \Theta(N)$, then remains bounded. Later, $\beta$ reaches $1/2$ after an additional time $T_2 = \Theta(N^2)$. Finally, $\alpha$ reaches $1/2$ after an additional time $T_3 = O(N^2)$. This gives the total times $T_\alpha = T_1 = \Theta(N)$, $T_\beta = T_1 + T_2 = \Theta(N) + \Theta(N^2) = \Theta(N^2)$, and $T_\gamma = T_1 + T_2 + T_3 = \Theta(N) + \Theta(N^2) + O(N^2) = \Theta(N^2)$. Each step is proven by appropriately bounding the gradient updates. We give the full proof below.

## B.2  SETUP

Recall the architecture and loss:

$$U = \left[\; X \;\middle|\; \sigma(XW^{(1)}X^\intercal)X \;\right] \qquad\qquad V = \left[\; U \;\middle|\; \sigma(UW^{(2)}U^\intercal)U \;\right]$$

$$z = V_{2N+1}\, W^{(3)} \qquad \mathcal{L} = \|y - z\|^2$$

where $[\,\cdot\mid\cdot\,]$ denotes matrix concatenation, and

$$W^{(1)} \in \mathbb{R}^{2D \times 2D} \qquad\qquad W^{(2)} \in \mathbb{R}^{4D \times 4D} \qquad\qquad W^{(3)} \in \mathbb{R}^{8D \times D}$$

$$U \in \mathbb{R}^{(2N+1) \times 4D} \qquad\qquad V \in \mathbb{R}^{(2N+1) \times 8D} \qquad\qquad z \in \mathbb{R}^D$$

$$X \in \mathbb{R}^{(2N+1) \times 2D} \qquad\qquad y \in \mathbb{R}^D$$

We use $\sigma$ to denote the softmax function with causal masking. We apply a causal mask that prevents a position from attending to itself, which is not a standard practice, but it greatly simplifies the proofs.

The data is generated as:

$$X_{2i-1} = \left[\; a_i \mid p_i \;\right] \qquad X_{2i} = \left[\; b_i \mid Mp_i \;\right] \qquad \forall i \in \{1, \dots, N\}$$

$$X_{2N+1} = \left[\; a_q \mid 0 \;\right] \qquad y = b_q$$

where

$$a_i,\; b_i,\; p_i \;\in\; \mathbb{R}^D \qquad q \in \{1, 2, \dots, N\} \qquad M = \begin{bmatrix} \mathbf{0} & I \\ \hline I & \mathbf{0} \end{bmatrix}$$

## B.3  LOSS FUNCTION

We begin by deriving a closed-form expression of the loss in terms of the three parameters.

The orthonormal inputs give us the following attention scores in the first layer:

$$(XW^{(1)}X^\top)_{ij} = \begin{cases} \alpha & i = 2k,\; j = i - 1 \\ 0 & \text{otherwise} \end{cases}$$

Applying the softmax attention with causal masking gives us:

$$\sigma(XW^{(1)}X^\top)_{ij} = \begin{cases} \dfrac{e^\alpha}{i - 2 + e^\alpha} & i = 2k,\; j = i - 1 \\[2mm] \dfrac{1}{i - 2 + e^\alpha} & i = 2k,\; j \neq i - 1 \\[2mm] \dfrac{1}{i - 1} & i = 2k + 1 \end{cases}$$

From Assump. 8, the target label is the last element in the sequence, following immediately after the queried item. This means that only the target label will contain the queried item after the first layer. Therefore, the target label will be the only position attended by the query:

$$(UW^{(2)}U^\top)_{2N+1,\,i} = \begin{cases} \beta \frac{e^\alpha}{2N-2+e^\alpha} & i = 2N \\ 0 & \text{otherwise} \end{cases}$$

Applying the softmax attention gives:

$$\sigma(UW^{(2)}U^\top)_{2N+1,\,i} = \begin{cases} \frac{s}{s+2N-1} & i = 2N \\ \frac{1}{s+2N-1} & \text{otherwise} \end{cases}$$

where $s = e^{\beta \frac{e^\alpha}{2N-2+e^\alpha}}$.

Applying the output projection layer will give us:

$$z = \frac{\gamma}{s + 2N - 1}\left( s\,b_N + a_N + \sum_{i=1}^{N-1} \big(a_i + b_i\big) \right)$$

The final loss will be:

$$\mathcal{L} = \|z - b_i\|^2 = \|z\|^2 - 2\,z^\top b_i + \|b_i\|^2$$
$$= \gamma^2 \frac{s^2 + 2N - 1}{(s + 2N - 1)^2} - 2\gamma \frac{s}{s + 2N - 1} + 1$$

where $s = e^{\beta \frac{e^\alpha}{2N-2+e^\alpha}}$.

Note that as long as inputs are orthonormal and the target label is in the last position, the loss only depends on $\alpha, \beta, \gamma$, and $N$. Any distribution over orthonormal inputs will give the same expected loss.

### B.4  LOSS GRADIENT

We now proceed to compute the partial derivatives of the loss function with respect to each of the three parameters.

#### B.4.1  AUXILIARY DEFINITIONS

$$G = e^\alpha + 2N - 2, \qquad\qquad\qquad F = 2N - 1,$$

$$s = \exp\!\Big(\frac{\beta\,e^\alpha}{G}\Big), \qquad\qquad\qquad r = s + F,$$

$$\mathcal{L} = \frac{\gamma^2\,(s^2 + F)}{r^2} \;-\; 2\gamma\,\frac{s}{r} \;+\; 1.$$

#### B.4.2  PARTIAL DERIVATIVE W.R.T. $\gamma$

$$\frac{\partial \mathcal{L}}{\partial \gamma} = \frac{\partial}{\partial \gamma}\Big( \tfrac{\gamma^2(s^2+F)}{r^2} - 2\gamma\,\tfrac{s}{r} + 1 \Big)$$

$$= 2\gamma\,\frac{s^2 + F}{r^2} \;-\; 2\,\frac{s}{r}$$

### B.4.3 PARTIAL DERIVATIVE W.R.T. $s$

$$\frac{\partial \mathcal{L}}{\partial s} = \frac{\partial}{\partial s}\left(\frac{\gamma^2(s^2+F)}{r^2}\right) - 2\gamma \frac{\partial}{\partial s}\left(\frac{s}{r}\right)$$

$$= \gamma^2 \frac{2s\,r^2 - (s^2+F)\,2r\,\frac{\partial r}{\partial s}}{r^4} - 2\gamma \frac{r - s\,\frac{\partial r}{\partial s}}{r^2}$$

But since $\partial r/\partial s = 1$,

$$\frac{\partial \mathcal{L}}{\partial s} = \frac{2\gamma^2 s\,r - 2\gamma^2(s^2+F)}{r^3} - 2\gamma\frac{r-s}{r^2}$$

$$= 2F\left(\frac{\gamma^2(s-1)}{r^3} - \frac{\gamma}{r^2}\right)$$

### B.4.4 DERIVATIVES OF $s$

$$s = \exp\left(\frac{\beta\,e^\alpha}{G}\right) \quad\Longrightarrow\quad \begin{cases} \dfrac{\partial s}{\partial \alpha} = s\,\dfrac{\partial}{\partial \alpha}\left(\dfrac{\beta\,e^\alpha}{G}\right) = s\,\dfrac{\beta\,e^\alpha(G-e^\alpha)}{G^2} = s\,\dfrac{2(N-1)\,\beta\,e^\alpha}{G^2} \\[2ex] \dfrac{\partial s}{\partial \beta} = s\,\dfrac{\partial}{\partial \beta}\left(\dfrac{\beta\,e^\alpha}{G}\right) = s\,\dfrac{e^\alpha}{G} \end{cases}$$

### B.4.5 APPLYING THE CHAIN-RULE RESULTS

$$\frac{\partial \mathcal{L}}{\partial \alpha} = \frac{\partial \mathcal{L}}{\partial s}\frac{\partial s}{\partial \alpha} = 2F\left(\frac{\gamma^2(s-1)}{r^3} - \frac{\gamma}{r^2}\right) \times s\,\frac{2(N-1)\,\beta\,e^\alpha}{G^2}$$

$$= \frac{4\,\beta\,(N-1)\,F\,s\,e^\alpha}{G^2}\left(\frac{\gamma^2(s-1)}{r^3} - \frac{\gamma}{r^2}\right)$$

$$\frac{\partial \mathcal{L}}{\partial \beta} = \frac{\partial \mathcal{L}}{\partial s}\frac{\partial s}{\partial \beta} = 2F\left(\frac{\gamma^2(s-1)}{r^3} - \frac{\gamma}{r^2}\right) \times s\,\frac{e^\alpha}{G}$$

$$= \frac{2\,F\,s\,e^\alpha}{G}\left(\frac{\gamma^2(s-1)}{r^3} - \frac{\gamma}{r^2}\right)$$

### B.4.6 FINAL RESULTS

$$\frac{\partial \mathcal{L}}{\partial \alpha} = \frac{4\,\beta\,(N-1)\,F\,s\,e^\alpha}{G^2}\left(\frac{\gamma^2(s-1)}{r^3} - \frac{\gamma}{r^2}\right)$$

$$\frac{\partial \mathcal{L}}{\partial \beta} = \frac{2\,F\,s\,e^\alpha}{G}\left(\frac{\gamma^2(s-1)}{r^3} - \frac{\gamma}{r^2}\right)$$

$$\frac{\partial \mathcal{L}}{\partial \gamma} = 2\gamma\frac{s^2+F}{r^2} - 2\frac{s}{r}$$

### B.4.7 VERIFICATION

We verify the correctness of the previous results using automated symbolic differentiation with the *SymPy* library. The code is provided with this paper.

### B.5 EMERGENCE OF IN-CONTEXT LEARNING

Combining the previously obtained loss derivatives with the zero initialization, we obtain the full set of constraints that determine our training trajectory:

$$\alpha(0) = \beta(0) = \gamma(0) = 0$$

$$\frac{\partial \alpha}{\partial t} = \frac{2\beta(2N-2)(2N-1)s\,e^\alpha}{(e^\alpha + 2N - 2)^2} \left( \frac{\gamma}{(s + 2N - 1)^2} - \frac{\gamma^2(s-1)}{(s + 2N - 1)^3} \right)$$

$$\frac{\partial \beta}{\partial t} = \frac{2(2N-1)s\,e^\alpha}{e^\alpha + 2N - 2} \left( \frac{\gamma}{(s + 2N - 1)^2} - \frac{\gamma^2(s-1)}{(s + 2N - 1)^3} \right)$$

$$\frac{\partial \gamma}{\partial t} = 2\frac{s}{s + 2N - 1} - 2\gamma \frac{s^2 + 2N - 1}{(s + 2N - 1)^2}$$

where $s = \exp\left(\beta \frac{e^\alpha}{e^\alpha + 2N - 2}\right)$.

We are interested in the first time $t_{\text{ICL}}$ when all three parameters are greater than $1/2$. As we show below, the parameters always reach this value in a specific order: first $\gamma$, then $\beta$, and finally $\alpha$.

We find the total time by breaking it down into three different times, one for each parameter:

$$t_{\text{ICL}} = T_1 + T_2 + T_3$$

We show that $\gamma$ emerges in $T_1 = \Theta(N)$, $\beta$ emerges after another $T_2 = \Theta(N^2)$, and finally $\alpha$ emerges after another $T_3 = O(N^2)$. This gives the total time:

$$t_{\text{ICL}} = \Theta(N) + \Theta(N^2) + O(N^2) = \Theta(N^2)$$

### B.6 EMERGENCE OF $\gamma$ IN $T_1 = \Theta(N)$

We start in the regime $0 \le \alpha, \beta, \gamma < \frac{1}{2}$. We show that $\gamma$ is the first to leave this regime at a time $T_1 = O(N)$.

#### B.6.1 DYNAMICS OF $\gamma$

Using $\alpha, \beta < \frac{1}{2}$, we get:

$$s = \exp\left(\beta \frac{e^\alpha}{e^\alpha + 2N - 2}\right) = 1 + O(1/N)$$

Using $\gamma < \frac{1}{2}$, we get:

$$\begin{aligned}
\frac{\partial \gamma}{\partial t} &= 2\frac{s}{s + 2N - 1} - 2\gamma \frac{s^2 + 2N - 1}{(s + 2N - 1)^2} \\
&\ge 2\frac{s}{s + 2N - 1} - \frac{s^2 + 2N - 1}{(s + 2N - 1)^2} \\
&\ge 2\frac{1 + O(1/N)}{2N + O(1/N)} - \frac{2N + O(1/N)}{(2N + O(1/N))^2} \\
&\ge \frac{1 + O(1/N)}{N} - \frac{2N + O(1/N)}{4N^2} \\
&\ge \frac{1}{2N} + O(1/N^2)
\end{aligned}$$

$$\frac{\partial \gamma}{\partial t} = 2 \frac{s}{s + 2N - 1} - 2\gamma \frac{s^2 + 2N - 1}{(s + 2N - 1)^2}$$

$$\leq 2 \frac{s}{s + 2N - 1}$$

$$\leq 2 \frac{1 + O(1/N)}{2N + O(1/N)}$$

$$\leq \frac{1}{N} + O(1/N^2)$$

This gives us

$$\frac{\partial \gamma}{\partial t} = \Theta(1/N)$$

Integrating over time, we obtain:

$$\gamma(T_1) = \int_0^{T_1} \frac{\partial \gamma}{\partial t} dt = T_1 \Theta(1/N)$$

Since $\gamma(T_1) = 1/2$, we get that $T_1 = \Theta(N)$.

### B.6.2   DYNAMICS OF $\alpha$ AND $\beta$

We are left to show that the condition $\alpha, \beta < \frac{1}{2}$ holds until $T_1$.

$$\frac{\partial \alpha}{\partial t} = \underbrace{\frac{2\beta (2N - 2)(2N - 1) s e^{\alpha}}{(e^{\alpha} + 2N - 2)^2}}_{O(1)} \left( \underbrace{\frac{\gamma}{(s + 2N - 1)^2}}_{O(1/N^2)} - \underbrace{\frac{\gamma^2(s - 1)}{(s + 2N - 1)^3}}_{O(1/N^4)} \right)$$

$$= O(1/N^2)$$

$$\frac{\partial \beta}{\partial t} = \underbrace{\frac{2(2N - 1) s e^{\alpha}}{e^{\alpha} + 2N - 2}}_{O(1)} \left( \underbrace{\frac{\gamma}{(s + 2N - 1)^2}}_{O(1/N^2)} - \underbrace{\frac{\gamma^2(s - 1)}{(s + 2N - 1)^3}}_{O(1/N^4)} \right)$$

$$= O(1/N^2)$$

Integrating over time, we get $\alpha(T_1) = O(1/N)$ and $\beta(T_1) = O(1/N)$. Therefore, for large enough $N$, it is guaranteed that $\alpha$ and $\beta$ will not reach $1/2$ by the time that $\gamma$ does.

### B.6.3   NON-NEGATIVITY

For completeness, we also show that parameters are always increasing within this regime, which guarantees that they will never become negative:

$$\frac{\partial \alpha}{\partial t} = \underbrace{\frac{2\,\beta\,(2N-2)\,(2N-1)\,s\,e^{\alpha}\gamma}{(e^{\alpha}+2N-2)^2}}_{\geq 0} \left( \underbrace{\frac{1}{(s+2N-1)^2}}_{\substack{\Theta(1/N^2)\\ \geq 0}} - \underbrace{\frac{\gamma(s-1)}{(s+2N-1)^3}}_{O(1/N^4)} \right) \geq 0$$

$$\frac{\partial \beta}{\partial t} = \underbrace{\frac{2\,(2N-1)\,s\,e^{\alpha}\gamma}{e^{\alpha}+2N-2}}_{\geq 0} \left( \underbrace{\frac{1}{(s+2N-1)^2}}_{\substack{\Theta(1/N^2)\\ \geq 0}} - \underbrace{\frac{\gamma(s-1)}{(s+2N-1)^3}}_{O(1/N^4)} \right) \geq 0$$

### B.7   EMERGENCE OF $\beta$ AFTER $T_2 = \Theta(N^2)$

We have now entered a new regime where $0 \leq \alpha, \beta \leq 1/2$ and $1/2 \leq \gamma \leq 3/2$. We will show that $\beta$ is the first to leave this regime after an additional time $T_2 = \Theta(N^2)$.

#### B.7.1   BOUNDING $\gamma$

We begin by showing that $\gamma$ remains bounded below $3/2$. We show that $\partial \gamma / \partial t$ would be negative at $\gamma = 3/2$, which implies that $\gamma$ will never go above $3/2$. We use the fact that $s = 1 + O(1/N)$ whenever $\alpha, \beta = O(1)$:

$$\begin{aligned}
\frac{\partial \gamma}{\partial t} &= 2\,\frac{s}{s+2N-1} - 2\gamma\,\frac{s^2+2N-1}{(s+2N-1)^2} \\
&= 2\,\frac{s}{s+2N-1} - 3\,\frac{s^2+2N-1}{(s+2N-1)^2} \\
&= 2\,\frac{1+O(1/N)}{2N+O(1/N)} - 3\,\frac{2N+O(1/N)}{(2N+O(1/N))^2} \\
&= \frac{1+O(1/N)}{N} - \frac{3N+O(1/N)}{2N^2} \\
&= -\frac{1}{2N} + O(1/N^2) \\
&< 0
\end{aligned}$$

#### B.7.2   DYNAMICS OF $\beta$

Applying the fact that $\gamma = \Theta(1)$ and $s = 1 + O(1/N) = \Theta(1)$ gives us:

$$\frac{\partial \beta}{\partial t} = \underbrace{\frac{2\,(2N-1)\,s\,e^{\alpha}\gamma}{e^{\alpha}+2N-2}}_{\Theta(1)} \left( \underbrace{\frac{1}{(s+2N-1)^2}}_{\Theta(1/N^2)} - \underbrace{\frac{\gamma(s-1)}{(s+2N-1)^3}}_{O(1/N^4)} \right) = \Theta(1/N^2)$$

By integrating, we obtain the value of $\beta$ after $T_2$:

$$\beta(T_1+T_2) = \beta(T_1) + \int_{T_1}^{T1+T_2} \frac{\partial \beta}{\partial t}\,dt = O(1/N) + T_2\,\Theta(1/N^2)$$

This gives us that $T_2\,\Theta(1/N^2) = 1/2$, which implies that $T_2 = \Theta(N^2)$.

### B.7.3 DYNAMICS OF $\alpha$

For completeness, we must establish that $\alpha$ does not become greater than $1/2$ before $\beta$. This comes from the fact that $\beta$ is always increasing at a faster rate than $\alpha$ in this regime:

$$\frac{\partial \alpha}{\partial t} = \underbrace{\frac{\beta(2N-2)}{e^\alpha + 2N - 2}}_{<1} \frac{\partial \beta}{\partial t}$$

### B.8 EMERGENCE OF $\alpha$ IN $T_3 = O(N^2)$

We have entered our last regime, which we define using the constraints $0 \le \alpha \le 1/2$, $1/2 \le \gamma \le 3/2$, and $1/2 \le \beta \le 20$.

We know from before that $\gamma$ remains constrained when $\alpha, \beta = \Theta(1)$. We are left to prove that $\alpha$ becomes greater than $1/2$ in a time $T_3 = O(N^2)$ and it does so before $\beta$ becomes greater than the value 20 (chosen arbitrarily to simplify the proofs).

### B.8.1 DYNAMICS OF $\alpha$

We establish an upper bound on $T_3$ using a lower bound on $\partial \alpha / \partial t$:

$$\frac{\partial \alpha}{\partial t} = 2\gamma\beta\, e^\alpha \underbrace{\frac{(2N-2)\,(2N-1)\,s}{(e^\alpha + 2N - 2)^2}}_{1+O(1/N)} \left( \underbrace{\frac{1}{(s + 2N - 1)^2}}_{1/(4N^2)+O(1/N^3)} - \underbrace{\frac{\gamma(s-1)}{(s + 2N - 1)^3}}_{O(1/N^4)} \right)$$

$$> \frac{1}{8N^2} + O\left(\frac{1}{N^3}\right)$$

Integrating over time gives:

$$\alpha(T_1 + T_2 + T_3) = \alpha(T_1 + T_2) + \int_{T_1 + T_2}^{T_1 + T_2 + T_3} \frac{\partial \alpha}{\partial t} dt$$

$$> T_3 \left( \frac{1}{8N^2} + O\left(1/N^3\right) \right)$$

Applying that $\alpha(T_1 + T_2 + T_3) = 1/2$ gives us $T_3 < 4N^2 + O(1/N) = O(N^2)$.

### B.8.2 DYNAMICS OF $\beta$

Finally, we must show that $\beta$ does not reach 20 during $T_3$. We achieve this using an upper bound on $\partial \beta / \partial t$:

$$\frac{\partial \beta}{\partial t} = 2\, e^\alpha\, \gamma \underbrace{\frac{(2N-1)\,s}{e^\alpha + 2N - 2}}_{1+O(1/N)} \left( \underbrace{\frac{1}{(s + 2N - 1)^2}}_{1/(4N^2)+O(1/N^3)} - \underbrace{\frac{\gamma(s-1)}{(s + 2N - 1)^3}}_{O(1/N^4)} \right)$$

$$< \frac{3\sqrt{e}}{4N^2} + O(1/N^3)$$

Integrating over time gives:

$$\beta(T_1 + T_2 + T_3) \;=\; \beta(T_1 + T_2) \;+\; \int_{T_1 + T_2}^{T1 + T_2 + T_3} \frac{\partial \beta}{\partial t} dt$$

$$< \;\; \frac{1}{2} \;+\; T_3 \left( \frac{3\sqrt{e}}{4N^2} + O\left(1/N^3\right) \right)$$

$$< \;\; \frac{1}{2} \;+\; \left( 4N^2 + O\left(1/N\right) \right) \left( \frac{3\sqrt{e}}{4N^2} + O\left(1/N^3\right) \right)$$

$$< \;\; \frac{1}{2} \;+\; 3\sqrt{e} + O(1/N)$$

$$< \;\; 5.45 + O(1/N)$$

$$< \;\; 20$$

## C  WEIGHTS DURING TRAINING

We confirm our theoretical result by visualizing the weights during standard training with stochastic gradient descent. We use learning rate $= 1$ and batch size $B = 512$.

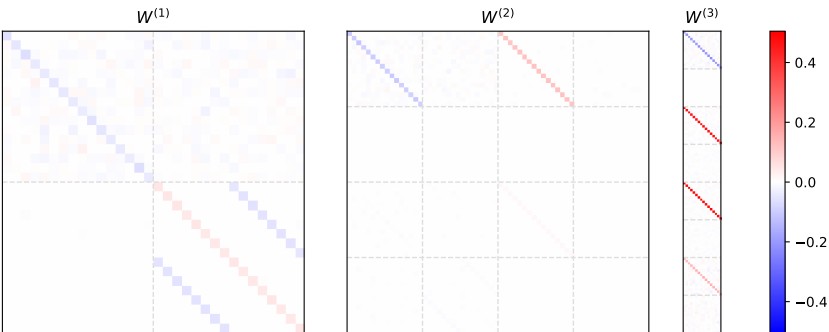

Figure 6: Model weights after 100 training steps with $D = 16$ and $N = 4$.

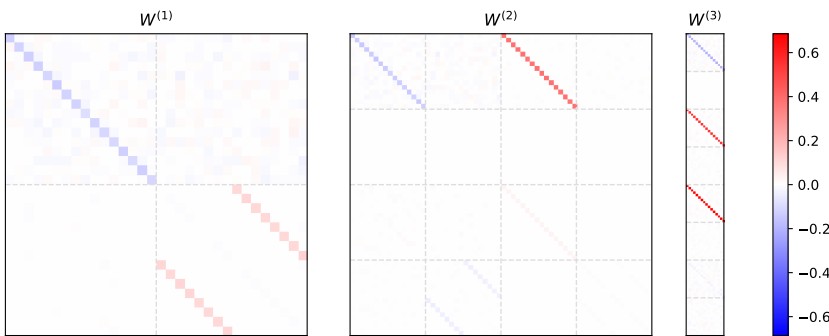

Figure 7: Model weights after 200 training steps with $D = 16$ and $N = 4$.

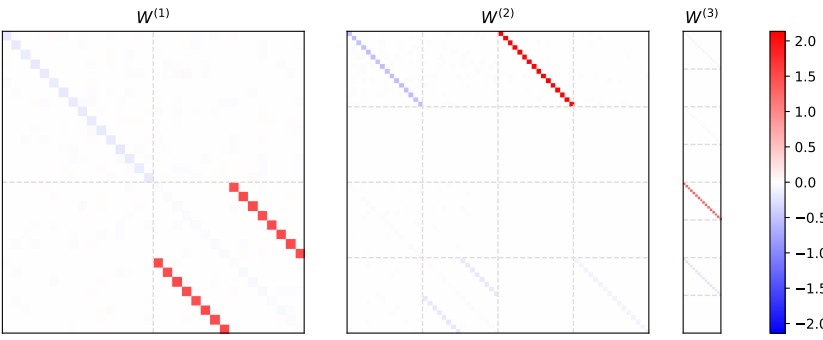

Figure 8: Model weights after 400 training steps with $D = 16$ and $N = 4$.

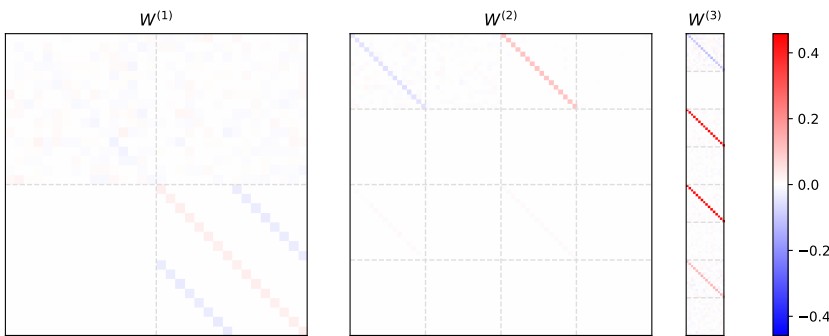

Figure 9: Model weights after 200 training steps with $D = 16$ and $N = 8$.

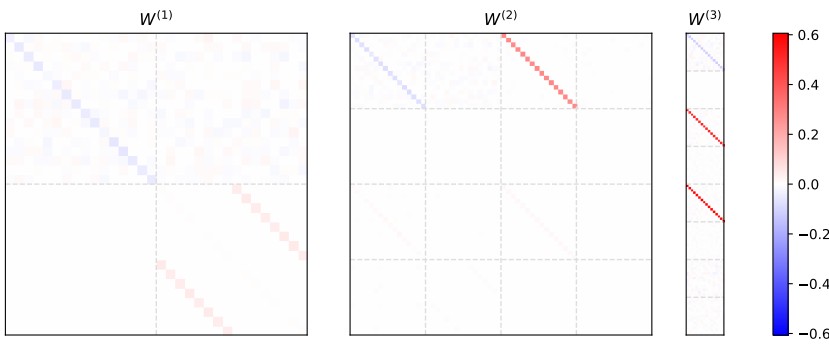

Figure 10: Model weights after 400 training steps with $D = 16$ and $N = 8$.

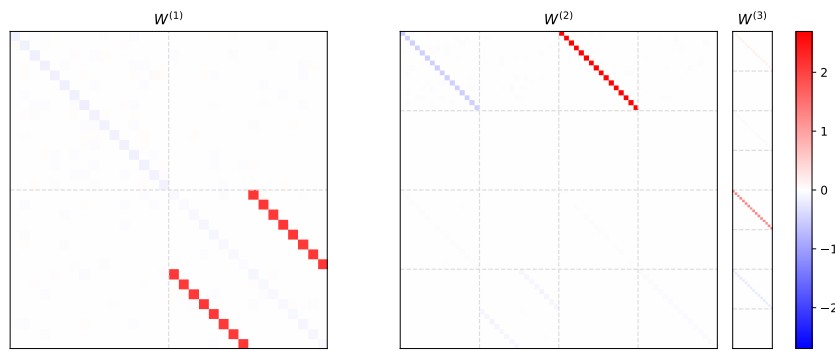

Figure 11: Model weights after 800 training steps with $D = 16$ and $N = 8$.

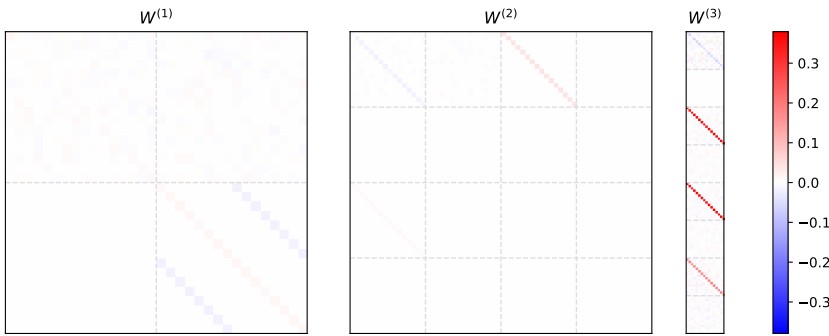

Figure 12: Model weights after 250 training steps with $D = 16$ and $N = 16$.

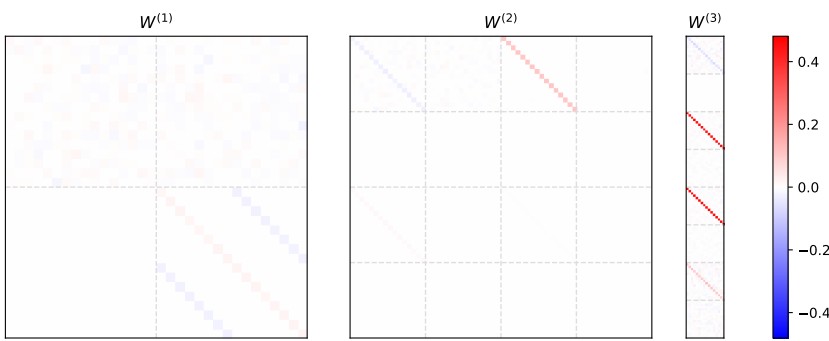

Figure 13: Model weights after 500 training steps with $D = 16$ and $N = 16$.

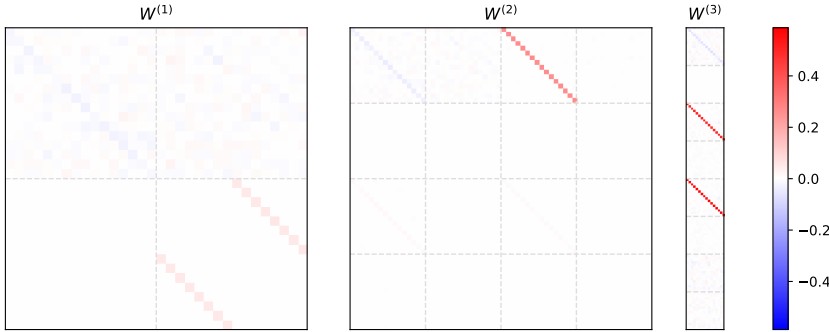

Figure 14: Model weights after 1000 training steps with $D = 16$ and $N = 16$.

# D TRAINING DYNAMICS

As in the main paper, we visualize the pseudo-parameters and loss during standard training, as well as when training only $\alpha_3$, $\beta_2$, and $\gamma_3$. We use $D = 32$, $N = 16$, learning rate $\lambda = 1$, and batch size $B = 256$. We determine the value of each pseudo-parameter by measuring the magnitude of the parameter vector along the corresponding component.

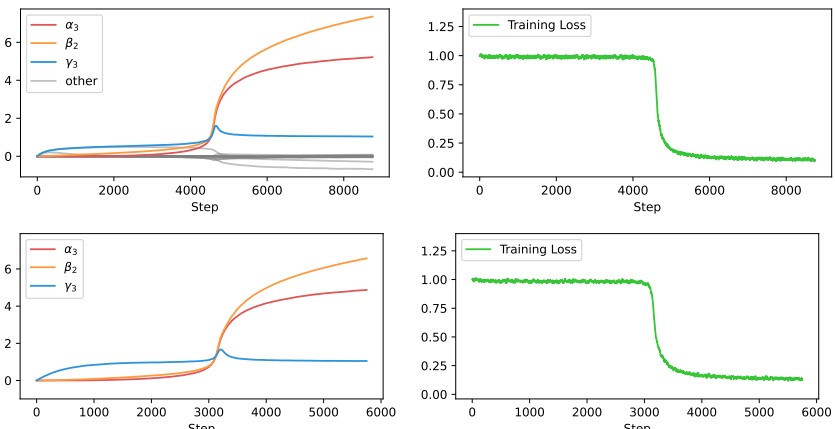

Figure 15: The pseudo-parameters and training loss during training with $D = 16$ and $N = 32$. *Top.* Standard training. *Bottom.* Training only $\alpha_3$, $\beta_2$, and $\gamma_3$.

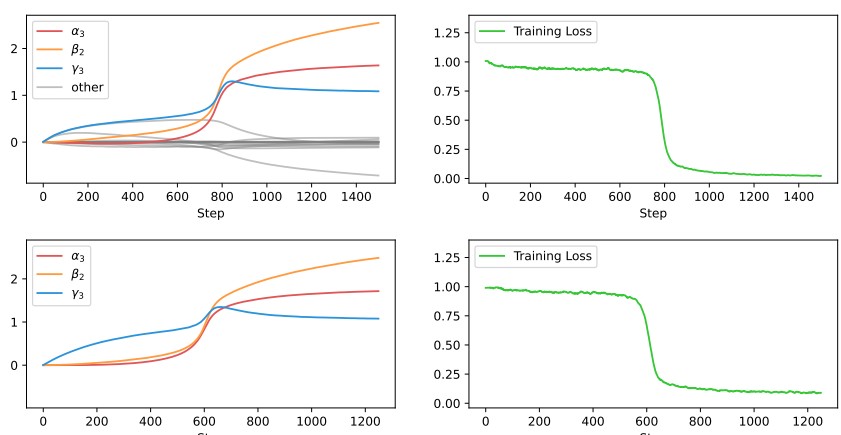

Figure 16: The pseudo-parameters and training loss during training with $D = 32$ and $N = 8$. *Top.* Standard training. *Bottom.* Training only $\alpha_3$, $\beta_2$, and $\gamma_3$.

# E    TRAINING DETAILS FOR SECTION 2

We use token and positional embeddings with a vocabulary size of 32, a block size of 32, and an embedding dimension of 2048. Since we have only one head per layer, the head dimension is also 2048. We do not use normalization or weight tying. Following standard practice, we train with AdamW (Loshchilov & Hutter, 2017) with learning rate 0.001 and weight decay 0.01. We train for 300 steps with 512 sequences per step. Every sequence has length 17 (8 item-label pairs and one query item) and is placed at a random position in the block. We generate new random sequences for every gradient step as follows: we choose 16 distinct tokens from our vocabulary and group them in item-label pairs; we choose one of the items to be the query; we use the corresponding label as the target output. We use the negative log-likelihood loss.

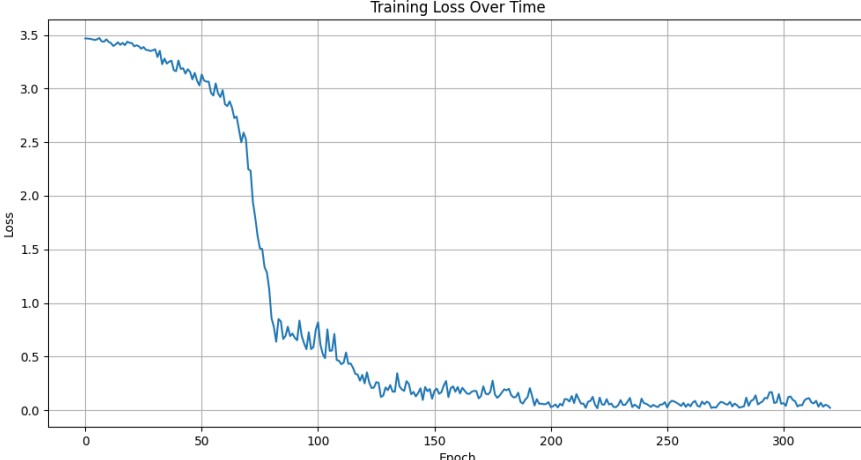

Figure 17: Training loss for the transformer used in Section 2. Note that every batch is generated independently, hence the training loss is also a test loss.

# F    TRAINING DETAILS FOR SECTION 6

We empirically validate our theoretical results by measuring the emergence times for different values of $N$. We find that emergence times are in accordance with theoretical predictions. Results are plotted in Fig. 5. We use $D = 256$, $B = 64$, $\lambda = 100$. Following our theoretical assumptions, we use orthonormal inputs, zero initialization, and $q = N$. We constrain the parameters to the 3-dimensional space spanned by $\boldsymbol{\alpha}_3, \boldsymbol{\beta}_2$, and $\boldsymbol{\gamma}_3$. Unlike our theory, we use a threshold of $0.1$ for $\boldsymbol{\alpha}_3$ and $\boldsymbol{\beta}_2$, (rather than $0.5$) to better highlight their separation.

