# OpenReview forum: "On the Emergence of Induction Heads for In-Context Learning"
_ICLR.cc/2026/Conference — ICLR 2026 Conference Withdrawn Submission_

### Official Review · Reviewer_YZwk · 2025-10-24

**Soundness:** 2
**Presentation:** 3
**Contribution:** 2
**Rating:** 6
**Confidence:** 2

**Summary:**

This paper investigates the training dynamics that lead to the emergence of "induction heads," a two-head mechanism known to be critical for in-context learning (ICL) in transformers. The work proceeds in two parts. First, the authors empirically analyze a standard two-layer, attention-only transformer trained on a simple ICL task, observing that the learned weights implementing the induction head have a clear, interpretable structure.

Second, to theoretically explain this emergence, the authors propose a *minimal* setup: a modified two-layer transformer with a "disentangled" (concatenated) residual stream and a simplified ICL task using Gaussian vector data. Under strong assumptions—most notably zero initialization, population loss, and isotropic data—they formally prove (Theorem 1) that the model's training dynamics are constrained to a 19-dimensional subspace. They then empirically show that only 3 of these 19 dimensions are relevant for forming the induction head. By further simplifying the assumptions (e.g., to orthonormal inputs) and studying the dynamics within this 3-dimensional subspace, they prove (Theorem 2) that the parameters emerge in a specific sequence and that the total time for the induction head to emerge ($t_{ICL}$) scales quadratically with the context length $N$, i.e., $t_{ICL} = \Theta(N^2)$.

**Strengths:**

1.  **Theoretical Rigor:** The paper provides a complete, end-to-end theoretical analysis of its minimal model, from subspace constraints (Theorem 1) to the final scaling laws (Theorem 2).
2.  **Clear Identification of Mechanism:** The empirical reduction from the 19-dimensional theoretical subspace to a 3-dimensional functional subspace ($\alpha_3, \beta_2, \gamma_3$) is a clean and insightful result.
3.  **Novel Scaling Law:** The derivation of the $t_{ICL} = \Theta(N^2)$ scaling law is a concrete, non-obvious, and testable theoretical prediction.
4.  **Intuitive Explanation:** The paper provides a clear, intuitive story (Sec 7) for *why* the parameters emerge in a specific sequence ($\gamma$ first, then $\beta$, then $\alpha$), which is a valuable piece of mechanistic insight.

**Weaknesses:**

1.  **Overly-Strong Assumptions:** The primary weakness is the reliance on a set of highly non-standard and simplifying assumptions, most critically **zero initialization**. This assumption is key to the symmetry argument of the main proof (Theorem 1) but is not representative of standard transformer training, thus limiting the applicability of all subsequent results.
2.  **Artificial Model and Data:** The theoretical analysis is not on a standard transformer, but a "disentangled" model. The data consists of Gaussian vectors, not tokens, and includes a strong, "built-in" rotational clue ($M p_i$) that may make the ICL task trivial.
3.  **Gap in Generalizability:** The paper does not bridge the gap between the toy model it analyzes (Sec 3-6) and the more standard transformer it interprets (Sec 2). It's unclear if the dynamics, emergence order, or scaling laws hold under standard (random) initialization, for additive residual streams, or on tokenized data. The finding that the dynamics are "self-contained" seems to apply only to the toy model itself.

**Questions:**

1.  The entire theoretical framework seems to depend on the perfect symmetry of the **zero initialization**. What happens if this assumption is relaxed to a standard, non-zero random initialization (e.g., Kaiming or Xavier)? Does the 19-dimensional constraint of Theorem 1 break immediately? If so, doesn't this imply the analysis is not applicable to standard training?
2.  How critical is the artificial $p_i \rightarrow M p_i$ positional cue for the task? Have the authors tested whether the 3-parameter induction head still emerges if this cue is removed (e.g., using the same $p_i$ for both item and label, or using no positional information for the label)?
3.  The $t_{ICL} = \Theta(N^2)$ scaling law is a strong prediction. Does this scaling match empirical observations of ICL emergence in *standard* transformer models (e.g., GPT-2), or is this scaling law an artifact of the paper's minimal setup?

---

### Official Review · Reviewer_3EwD · 2025-10-29

**Soundness:** 3
**Presentation:** 3
**Contribution:** 2
**Rating:** 4
**Confidence:** 4

**Summary:**

This paper uses a simple experimental set up to induce learning of induction heads and studies how the learning process unfolds.  The  authors are able to characterize the learning process in their set up mathematically.  They isolate three patterns (what they call pseudo parameters) in the structure of the attention weights and they give temporal bounds on when these patterns appear.  The study is restricted to a 2 attention layer only (no MLP) transformer model without normalisation.

**Strengths:**

The analysis of this minimal model is interesting and convincing.  Theorems 1 and 2 show what's going on in this little model on this induction head learning task.

**Weaknesses:**

For this reviewer, it worth remembering that we are examining the emergence of a particular behavior from a task that was designed to induce this behavior.  Thus the results have to be treated with caution.

First Olsson et al 2022 and Reddy (2023) give suggestive evidence that icl patterns with a certain learnable behavior (induction head behavior and with a certain loss across token indices).   But it's not terribly surprising that even very simple models can learn this copying behavior.
After that however matters still remain very speculative as to the link between induction heads and general icl. the increase in icl accuracy in Olsson is measured in loss across token indices and it's not clear how to correlate this with actual task predictions in icl.  Additionally, empirical work has shown that for slightly more complex tasks, the task of learning linear functions (Garg et al. 2022) the induction head mechanism of Olsson gives incorrect predictions (Naim and Asher2024).  This weakens the explanatory relevance of induction heads.

The authors write "Interestingly, the mechanism performed by these three parameters together corresponds exactly to an induction head."  The authors do show how this particular task is implemented in the attention mechanism from learning about induction heads and exactly what weights change in this learning process.  But this depends on the training task and some particular assumptions about data and initial set up of the model; what happens when we try to generalize away from these assumptions?  Outside of the simple task set up, we have no reason to believe that induction heads are formed in this way.
The connections to icl remain elusive.

Naim and Asher (2024) also have experiments on context length in icl with linear functions that might interest the authors.

**Questions:**

Please comment on how you see the explanatory relevance of induction heads in general for icl.  I believe it's important for seeing the import of this work.

Why is there no normalisation in your models?

typo: implement and induction head ---> implement an induction head

suggested bibliographical item: Omar Naim and Nicholas Asher. Re-examining learning linear functions in context. arXiv:2411.11465
2024

---

### Official Review · Reviewer_NpqX · 2025-11-01

**Soundness:** 3
**Presentation:** 3
**Contribution:** 3
**Rating:** 6
**Confidence:** 3

**Summary:**

This paper investigates the training dynamics that lead to the emergence of induction heads, a key two-layer mechanism responsible for in-context learning (ICL) in transformers. The authors first analyze a standard two-layer attention-only transformer to identify an interpretable weight structure for induction heads. To make theoretical analysis tractable, they then propose a minimal ICL task and a simplified, "disentangled" transformer architecture.

The primary theoretical contribution (Theorem 1) is a formal proof that under assumptions of zero initialization, population loss, and isotropic data, the training dynamics of this minimal model are constrained to a 19-dimensional parameter subspace. Empirically, the authors demonstrate that only 3 of these 19 dimensions are necessary for the ICL task, corresponding to the induction head mechanism.

The second main theoretical contribution (Theorem 2) analyzes the gradient flow dynamics within this 3-dimensional subspace. Under further, more restrictive assumptions (orthonormal inputs, query-last position), the authors prove that the parameters emerge in a fixed sequence and derive a tight asymptotic bound for the total emergence time of the induction head, showing it is quadratic in the context length $N$ ($t_{ICL} = \Theta(N^2)$).

**Strengths:**

1. Fundamental Problem: The paper tackles a core question in mechanistic interpretability: not just what circuits (like induction heads) exist, but how they are learned by gradient descent. Moving from a static to a dynamic analysis is a valuable contribution.

2. Novel Theoretical Result: Theorem 1, which proves that training dynamics are constrained to a 19-dimensional subspace, is a strong and elegant result. The proof technique using data isotropy and rotational symmetry is clever.

3. Concrete, Testable Prediction: The $\Theta(N^2)$ scaling law for emergence time is a precise, non-trivial prediction. This provides a formal basis for the empirical observation that ICL is data- and time-intensive to learn, linking context length directly to training time.

**Weaknesses:**

My primary concerns relate to the significant gap between the highly simplified model used for the theory and a standard transformer, as well as several restrictive assumptions and internal inconsistencies that call the generality of the main results into question.

1. Gap Between Standard and Minimal Architectures: The paper begins by motivating the problem with a "standard" attention-only transformer (§2, Fig. 2) but then immediately pivots to a highly artificial, minimal architecture for all theoretical proofs (§3). This model uses a "disentangled" residual stream (concatenation, not addition), merged key-query matrices, and no MLPs or LayerNorms. The claim that this is "equally powerful" (line 164) is insufficient. It is not at all clear that the learning dynamics (Theorems 1 & 2) derived for this minimal model have any bearing on the standard transformer from §2, let alone on practical, deep transformers. The bridge between the two is missing.

2. Highly Restrictive Theoretical Assumptions: The theoretical results stand on a fragile foundation of strong, simplifying assumptions. For Theorem 1, zero initialization is not a "reasonable approximation for small random initializations" (line 226); it is a precise, symmetry-inducing condition that is critical for the proof. Population loss ignores the stochasticity of SGD. For Theorem 2, the assumptions become even more severe. Orthonormal inputs ($D \ge 2N$) is a strong data constraint. Most critically, Assumption 8 (Query Last, $q=N$) is a massive simplification of the ICL task. It fixes the relative position of the query and target, which seems to remove much of the complexity of the task (i.e., finding the correct prior $A$ to copy $B$ from).

**Questions:**

1. Please clarify the contradiction regarding the query position $q$. Which experiments (Fig 4? Fig 5?) use a random $q$, and which use a fixed $q=N$? How does the $\Theta(N^2)$ scaling change if $q$ is uniformly random, as in the task's initial definition?

2. Can you provide more intuition on the role of the 16 "other" pseudo-parameters from Theorem 1? Why do they exist in the gradient-space if they are not needed for the task?

3. How do you justify that the dynamics of your highly simplified, non-residual, disentangled model in section 3 are representative of the dynamics in the standard (additive residual stream) transformer you present in section 2?

---

### Official Review · Reviewer_vB5A · 2025-11-03

**Soundness:** 2
**Presentation:** 2
**Contribution:** 2
**Rating:** 2
**Confidence:** 2

**Summary:**

The paper studies how induction heads emerge during in‑context learning (ICL). Using a minimal transformer with disentangled  residuals.
The authors show that, under strong  assumptions, training dynamics remain in a 19‑dimensional subspace and only three pseudo‑parameters (alpha, beta, gamma) matters.

**Strengths:**

Three‑parameter dynamics, order gamma->beta-> alpha and the quadratic scaling, are well‑motivated inside the minimal setup, with closed‑form losses and derivatives (Appendix B) and consistent empirical plots (Fig. 5).

**Weaknesses:**

The paper has over‑reliance on strong, non‑standard assumptions in Transformer.
The key results hinge on concatenated residuals, isotropy data, and even forbid self‑attention to the current position, which are explicitly not a standard practice.
Since the proofs and most experiments apply to disentangled residuals under isotropy and other constraints, the claims and title should explicitly limit scope; otherwise readers may infer general statements about standard transformers that are not actually established.

**Questions:**

Q. 1 Section 3.1 asserts that concatenation is “equivalent to a very large residual dimension” and makes “activations almost orthogonal,” but no quantitative evidence (e.g., subspace principal angles, cosine coherences, Gram off‑diagonals) is provided, nor a comparison against additive residuals.
Is it possible to quantify the the claim and compare concatenation vs addition?

Q.2 What happens if you allow self‑attention to the current position (dropping the non‑standard mask)?
 Which parts of Theorem 2’s gradient‑order arguments rely on that restriction?
Can you provide a dependency map of lemmas on this assumption?

Q3. The paper defines emergence at ">=0.5" (Def. 1) but uses "0.1" for alpha_3,beta_2  in Appendix F "to highlight separation", which is threshold inconsistency.
Could you show and explain the results under threshold ">=0.5"?

Q4. Can you scale up the experiment (D=D_H=2048, N_E=N_p=32 is very small) to larger vocab/positional sizes and longer contexts.

Q5. How do your results relate to the recent work that positional encodings can produce pseudo‑induction and break length generalization (e.g., Wang et al., and Rethinking Associative Memory Mechanism in Induction Head, COLM2025, arXiv:2412.11459)?
This question is related to Q. because the small sequence can easily produce pseudo‑induction with positional encodings.
You need to check the length generlization with long sequences.
The work suggested that positional encodings can induce pseudo‑induction behavior and fail length generalization.
The present setup may suffer similar artifacts.

Minor comments:
Appendix B.1’s summary should swaps T_alpha and T_gamma.
It states T_alpha=T_1 and T_gamma =T_1+T_2+T_3.

---

### Note · Authors · 2026-01-04

I have read and agree with the venue's withdrawal policy on behalf of myself and my co-authors.